# Interference-Aware K-Step Reachable Communication in Multi-Agent Reinforcement Learning

Ziyu Cheng[1*]                              23371427@buaa.edu.cn
Jinsheng Ren[2*†]                    renjinsheng@qiyuanlab.com
Jun Yang[2†]                         yangjun603@tsinghua.edu.cn
Zhouxian Jiang[2]                  jiangzhouxian@qiyuanlab.com
Chenzhihang Li[2]                     li_chenzhihang@bit.edu.cn
Rongye Shi[3]                             shirongye@buaa.edu.cn
Bin Liang[2]                           liangbin@qiyuanlab.com

[1]School of Software, Beihang University  [2]QiYuan Lab  [3]School of Artificial Intelligence, Beihang University

## Abstract

Effective communication is pivotal for addressing complex collaborative tasks in multi-agent reinforcement learning (MARL). Yet, limited communication bandwidth and dynamic, intricate environmental topologies present significant challenges in identifying high-value communication partners. Agents must consequently select collaborators under uncertainty, lacking a priori knowledge of which partners can deliver task-critical information. To this end, we propose Interference-Aware $K$-Step Reachable Communication (IA-KRC), a novel framework that enhances cooperation via two core components: (1) a $K$-Step reachability protocol that confines message passing to physically accessible neighbors, and (2) an interference-prediction module that optimizes partner choice by minimizing interference while maximizing utility. Compared to existing methods, IA-KRC enables substantially more persistent and efficient cooperation despite environmental interference. Comprehensive evaluations confirm that IA-KRC achieves superior performance compared to state-of-the-art baselines, while demonstrating enhanced robustness and scalability in complex topological and highly dynamic multi-agent scenarios.

## 1 Introduction

Multi-Agent Reinforcement Learning (MARL) enables collaborative decision-making through interactions between multiple agents and their environment. This paradigm has demonstrated significant potential in autonomous driving (Chen et al., 2025), game AI (Vinyals et al., 2019), UAV coordination (Liao et al., 2025), where effective inter-agent communication is crucial for successful cooperation. However, practical constraints in communication bandwidth and system scalability make fully-connected communication infeasible (Zhu et al., 2024), necessitating efficient distributed communication strategies (Hu et al., 2024). This raises a fundamental research question: How to identify the most valuable communication partners in complex multi-agent systems? Prior work (Siedler, 2021) shows that poor partner selection can not only diminish cooperation benefits but may actually degrade overall system performance. Moreover, real-world environments often possess complex topologies and exhibit highly dynamic group behaviors, further complicating the identification of high-value communication targets.

Existing approaches for selecting communication partners mainly rely on neighborhood-based constraints. Hüttenrauch et al. (2019); Jiang & Lu (2018) use Euclidean distance as the selection criterion, considering spatially nearby agents as potential communication partners. However, in complex environments, Euclidean distance often significantly overestimates actual reachability, resulting in inefficient communication. As

---

[1]*Equal contribution. †Corresponding author.

shown in Figure 1 (a), when obstacles are present, agents A and B may have a small Euclidean distance (blue dashed line) but be separated by a long traversable path (yellow arrow path), hindering effective cooperation. To overcome this limitation, other works have proposed visual perception-based approaches that establish communication links only between agents with direct line of sight (green region) (Chen et al., 2021; Baker et al., 2019) (Figure 1 (b)). While this method more accurately reflects physical connectivity, it remains limited in complex environments where agents may fail to detect nearby partners that are occluded from view.

Moreover, existing methods often overlook the interference caused by adversarial dynamics and interactions among agents, which can lead to ineffective cooperation even when agents are close neighbors. For instance, enemy attacks can create high-risk zones that severely disrupt the cooperation of friendly agents, thereby significantly increasing the cost of cooperation. As illustrated in Figure 1(c), the red high-risk zone severely disrupts communication between agents A and B, forces them to take a detour to cooperate, resulting in prohibitively high path transition costs. This observation underscores the critical importance of dynamic interference modeling for effective collaborator selection. While recent work by Naderializadeh et al. (2020) attempts to address this challenge using graph neural networks (GNNs) to infer cooperative structures and implicitly capture adversarial interference through end-to-end learning, such approaches typically exhibit limited scalability and performance degradation in large-scale multi-agent scenarios (Gogineni et al., 2023).

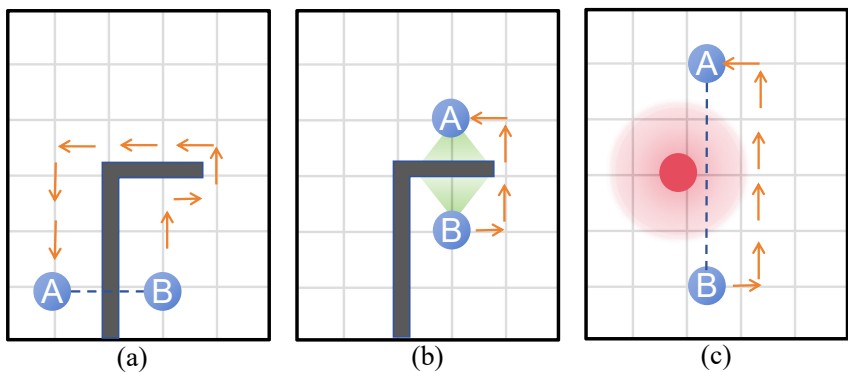

Figure 1: Limitations of common neighborhood constraints in MARL communication. (a) Euclidean distance may misrepresent actual accessibility: agents A and B appear close (blue dashed line) but are separated by a long traversable path (yellow arrow); (b) Vision-based constraints capture physical proximity better but miss reachable agents hidden from view; (c) Even with direct visibility, hostile interference (red region) can block cooperation, forcing detours (yellow arrows).

This paper posits that high-value communication should occur between agents that are both physically reachable and experience minimal interference, thereby enabling sustained and effective collaboration. To this end, we propose the Interference-Aware $K$-step Reachable Communication (IA-KRC), which consists of two modules as shown in Figure 2. **$K$-step Reachable Communication**: This module restricts communication to agents that are mutually reachable within $K$ movement steps. This definition considers agents' actual mobility capabilities in complex environments, providing a more accurate characterization of neighborhood relationships compared to Euclidean distance or line-of-sight metrics. **Interference-Prediction Module**: This component evaluates potential interference from agents beyond the $K$-step neighborhood, including adversarial interference and cooperative conflicts. By integrating these predictions, IA-KRC can further identify low-interference, high-value communication partners within the reachable domain. Furthermore, we develop an IA-KRC-based learning algorithm that efficiently identifies high-value communication and optimizes policy learning in an end-to-end framework.

We validate IA-KRC in challenging multi-agent combat scenarios constructed within the StarCraft Multi-Agent Challenge (SMACv2) framework Samvelyan et al. (2019), featuring dense obstacles and maze-like topologies. Under our self-play framework against strong baselines including CommFormer, Euclid, Vision, RL-Vision, MAPPO, and QMIX, IA-KRC achieves a win rate advantage of at least 4.58× and up to 31.56×.

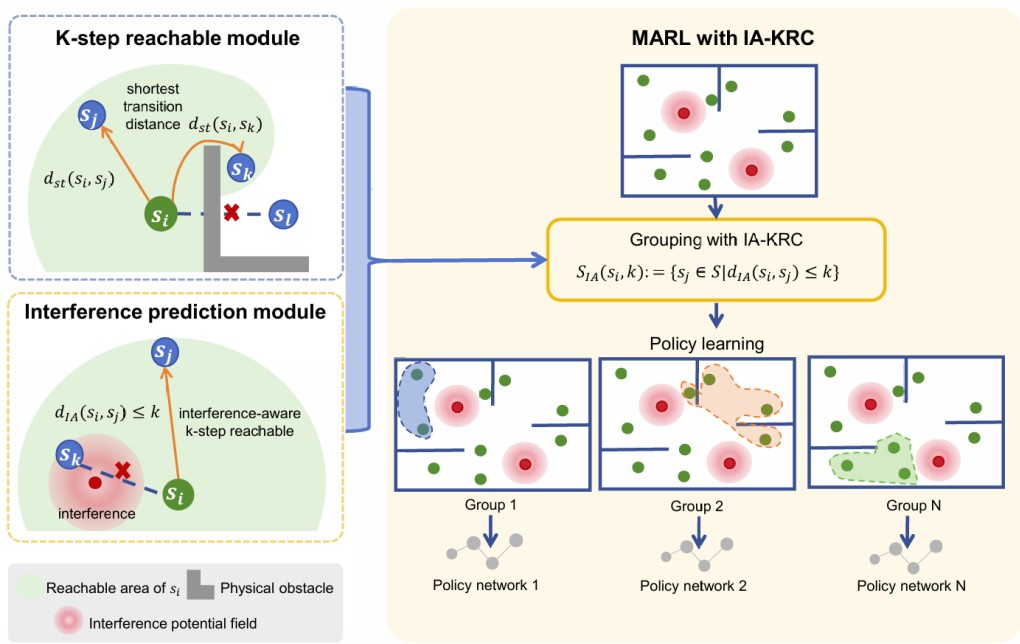

Figure 2: Overview of IA-KRC framework for MARL communication. The K-step Reachability Module restricts communication to agents within a physically reachable domain based on shortest transition distance. The Interference Prediction Module evaluates potential conflicts or adversarial effects, selecting low-interference partners. Combined, these modules enable dynamic grouping and robust leader–follower collaboration in complex environments.

This significant advantage stems from IA-KRC's ability to achieve sustained and effective cooperation in topologically complex and interference-laden environments.

## 2 Related Work

**Learning to communicate in MARL.** Effective communication is critical for collaboration in MARL. However, under bandwidth and scalability constraints, determining *whom to communicate with* becomes a central challenge.

Traditional metrics such as Euclidean distance or line-of-sight visibility often fail to accurately capture the constraints of communication relevant neighborhoods. Recent efforts learn communication topology end-to-end using attention mechanisms or GNNs Naderializadeh et al. (2020); Hu et al. (2024); Niu et al. (2021); Su et al. (2020), such as introducing personalized peer-to-peer communication via learned message routing policies Meng & Tan (2024). Although powerful in small-scale settings, these models lack explicit spatial priors (e.g., physical reachability), do not account for dynamic interference, and often scale poorly with the number of agents Christianos et al. (2021). Beyond spatial cues, some methods explore semantic-level grouping Sunehag et al. (2018), or employ self-supervised learning for message aggregation Guan et al. (2022), but these struggle with dynamic spatial feasibility in topologically constrained tasks. Furthermore, recent advances learn communication graphs via differentiable architecture search and graph–transformer hybrids Zhang et al. (2025); Inala et al. (2020), or adopt multi-level asynchronous communication for sequential coordination Ding et al. (2024), to obtain dynamic, sparse topologies with scalable routing. While these methods improve communication efficiency through learned selection, aggregation, or hierarchical policies, most operate on abstract feature spaces without explicitly grounding communication in physical environmental constraints, and consequently may be less effective when spatial accessibility and dynamic physical interference fundamentally constrain collaboration. Our work addresses these limitations by introducing a more faithful spatial foundation for communication learning: we explicitly model both multi-step physical

reachability and dynamic environmental interference as structural priors for communication, enabling robust partner selection in physically grounded, dynamic environments.

**The concept of $K$-Step reachability** is well-established in single-agent reinforcement learning, where it primarily serves to constrain the selection of subgoals Nachum et al. (2018); Vezhnevets et al. (2017). Our work represents the first application of reachability constraints within multi-agent systems, specifically for the purpose of selecting communication partners. A critical limitation of prior approaches is their difficulty in scaling to multi-agent settings. This is because the inherent interference in such environments—arising from teammate motion, opponent behaviors, and evolving policies Lowe et al. (2017)—means that relying solely on reachability is insufficient to guarantee sustained and effective cooperation. Addressing this gap, our paper introduces, for the first time, an interference-aware formulation of $K$-Step reachability. While some works have modeled influence or interference Jiang et al. (2020); Omidshafiei et al. (2022), these concepts have not been integrated with reachability to guide communication. Our approach closes this gap by jointly modeling physical reachability and dynamic interference, enabling robust partner selection in physically grounded, dynamic environments.

## 3 Preliminaries

In this work, we consider the cooperative MARL scenario formulated as a decentralized partially observable Markov decision process (Dec-POMDP) (Oliehoek et al., 2016), a common framework for modeling cooperation among multiple autonomous agents under uncertainty. Formally, a Dec-POMDP is described as a tuple $\langle \mathcal{S}, \mathcal{A}, \mathcal{O}, \mathcal{P}, r, \gamma, N \rangle$, where $\mathcal{S}$ represents the global state space shared by all agents, $\mathcal{A} = \mathcal{A}_1 \times \mathcal{A}_2 \times \cdots \times \mathcal{A}_N$ is the joint action space of $N$ agents, and $\mathcal{O}$ is the observation space available to each agent. Let $\mathcal{N} = \{1, \ldots, N\}$ denote the set of agents. The state transition probability $\mathcal{P}(s' \mid s, a) \colon \mathcal{S} \times \mathcal{A} \times \mathcal{S} \to [0, 1]$ specifies the probability of transitioning to state $s'$ from state $s$ given joint action $a$. The global reward function $r(s, a) \colon \mathcal{S} \times \mathcal{A} \to \mathbb{R}$ defines the common objective for all agents. The discount factor $\gamma \in [0, 1)$ determines the relative importance of future rewards. Each agent $i$ independently receives a local observation $o_i$ based on an observation function $O(s, i) \colon \mathcal{S} \times \{1, \ldots, N\} \to \mathcal{O}$.

To facilitate effective decentralized cooperation, agents are partitioned into cooperative groups, where each group $g \in G$ consists of a leader and several followers. Each group $g$ maintains a joint action-observation history $\boldsymbol{\tau}_g$ and learns a shared group policy $\pi_g(\mathbf{a}_g \mid \boldsymbol{\tau}_g) \colon \mathcal{T}^{|g|} \to \Delta(\mathcal{A}_g)$, where $\mathcal{T}^{|g|}$ denotes the space of joint histories for agents in group $g$, $\mathcal{A}_g$ is the joint action space of the group, and $\Delta(\mathcal{A}_g)$ is the probability simplex over $\mathcal{A}_g$. Our goal is to efficiently resolve leader election and follower assignment.

## 4 Method

The objective of IA-KRC is to facilitate sustained and effective cooperation by ensuring that communication occurs between agents that are not only physically reachable but also subject to minimal interference. As illustrated in Figure 2, the IA-KRC framework comprises two main components: the $K$-Step Reachability Module and the Interference Prediction Module. The former quantifies the physical path accessibility between agents, restricting communication to those in close proximity. The latter assesses the potential communication interference that other agents may impose on a target agent. By integrating these two modules, IA-KRC enables the selection of reliable, long-term cooperative partners within a neighborhood by filtering for low-interference candidates.

The principle of $K$-step reachability dictates that an agent must be able to reach the state of another agent within $K$ time steps. In this context, conventional proximity metrics like Euclidean distance are inadequate, as they often fail to capture the underlying topological structure of MDPs. To address this, we introduce the concept of the shortest transition distance as a more suitable metric for evaluating reachability between agents. In a stochastic MDP, the number of steps required for a transition is not deterministic. Consequently, we define shortest transition distance by minimizing the expected first-hitting time over the set of all possible policies, as formally presented in Definition 1.

**Definition 1.** Let $x_1, x_2 \in \mathcal{X}$ be two agent states. Then, shortest transition distance from $x_1$ to $x_2$ is defined as:

$$d_{st}(x_1, x_2) := \min_{\pi \in \Pi} \mathbb{E}[T_{x_1, x_2} | \pi] = \min_{\pi \in \Pi} \sum_{t=0}^{\infty} t P(T_{x_1, x_2} = t | \pi),$$

where $\Pi$ is the complete policy set and $T_{x_1, x_2}$ denotes the first hit time from $x_1$ to $x_2$.

Considering the topological complexity, we do not assume the environment is reversible; that is, $d_{st}(x_1, x_2)$ is not necessarily equal to $d_{st}(x_2, x_1)$. Therefore, shortest transition distance is a quasi-metric, as it does not satisfy symmetry. Shortest transition distance measures the time steps required to transition from state $x_1$ to $x_2$ in the most efficient manner, and it has been a subject of study in several works on single-agent reinforcement learning (Chebotarev & Deza, 2020; Zhang et al., 2020). However, in multi-agent environments, selecting communication partners based solely on this distance is insufficient. This is because actions from other agents can interfere with cooperation, even when two agents are in close proximity. For instance, cooperative agents working together to unlock mechanisms can reduce shortest transition distance, while adversarial agents may block critical pathways, rendering the shortest transition routes practically unreachable. To address this, our proposed Interference Prediction Module explicitly measures the cost of cooperation. Accordingly, the interference-aware shortest transition distance is defined as follows.

**Definition 2.** Let $C(T_{x_1, x_2} = t | \pi)$ be the cooperation cost incurred when the first-hitting time from $x_1$ to $x_2$ is $t$ under policy $\pi$. We assume $C(T_{x_1, x_2} = t | \pi) \geq 0$. Then, the interference-aware shortest transition distance is defined as:

$$d_{IA}(x_1, x_2) := \min_{\pi \in \Pi} \sum_{t=0}^{\infty} t P(T_{x_1, x_2} = t | \pi) \times C(T_{x_1, x_2} = t | \pi).$$

Based on the interference-aware shortest transition distance, we introduce the notion of interference-aware $K$-step reachable region to bound the communication scope among agents.

**Definition 3.** Let $x_1 \in \mathcal{X}$. Then, the interference-aware $K$-Step reachable region of $x_1$ is defined as:

$$\mathcal{S}_{IA}(x_1, K) := \{x_2 \in \mathcal{X} \mid d_{IA}(x_1, x_2) \leq K\}.$$

### 4.1 Computing $K$-step reachability using multi-layer map

Although the interference-aware reachable region has been formally defined, evaluating shortest transition distance $d_{st}(s_1, s_2)$ between any two agent remains non-trivial. In a non-stationary MARL environment, the distance calculated at time $t$ can become invalid at $t + n$: moving agents or obstacles may block the previously optimal path, forcing $d_{st}$ to change (see Figure 3a). Re-computing all pairwise distances every step is prohibitively expensive, while caching past results is unreliable because their validity quickly decays. Some prior works have attempted to utilize multi-layer perceptrons to record and store state neighborhood information (Paraskevopoulos et al., 2017). However, MLPs struggle to fit non-stationary data, rendering this approach ineffective in our context.

To avoid unnecessary global updates, our approach first detects where the environment has actually changed and then refreshes distances only for those local regions. Empirically, shortest transition distances evolve on distinct time scales: obstacle-induced changes are slow, environmental rule-based changes (e.g., door states as shown in Figure 3a) occur at a moderate rate, whereas agent state changes and adversarial interference alter transition costs almost instantly. We therefore introduce a multi-layer map (Figure 3b) that separates information by its rate of change, in which each layer records elements that change on different time scales, thereby allowing us to quickly localize the environmental regions where changes have occurred.

The multi-layer map serves as an abstract representation of the multi-agent environment, comprising three mutually decoupled layers. The Geometric Layer stores static elements and extremely slowly changing dynamic elements, updated through agent observations. The Regulation Layer stores environmental rule-based information, such as the opening and closing of doors and changes in traffic lights in real-world scenarios. This rule-based information evolves at a moderate rate and is updated based on the actual transitions of

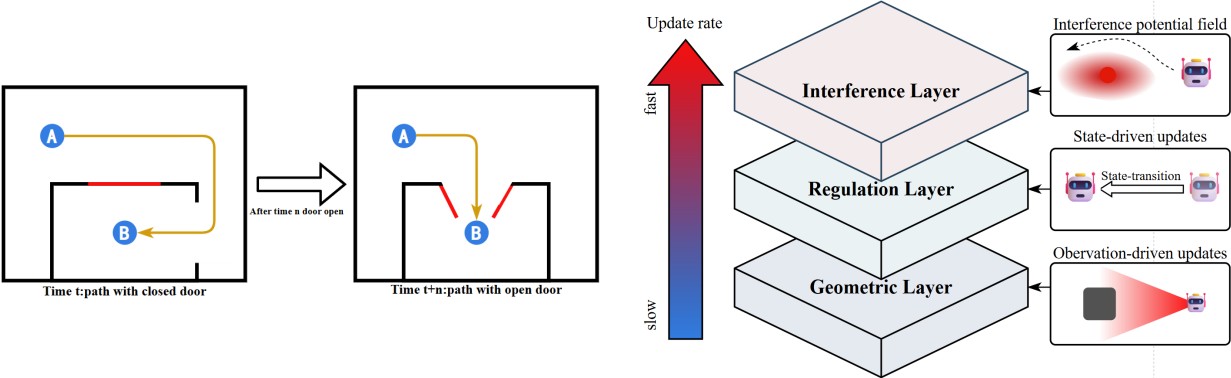

(a) Impact of time-varying regulation rules on shortest transition distance. Red line: door state changing over time; yellow arrow: shortest transition path.

(b) Illustration of the multi-layer map used to compute $K$-Step reachability.

Figure 3: Illustration of the time-varying regulation (a) and the multi-layer map structure (b).

agents within the environment. The Interference Layer stores adversarial interference information of entities, dynamically generating real-time interference potential fields from the stored information to precisely track these most rapidly changing attributes. By monitoring the confidence levels of all layers and updating information below confidence thresholds, the map achieves asynchronous updates. **Rather than replicating the full environment, this abstraction discretizes and models only the essential dynamic factors, enabling efficient tracking of non-stationary elements**. Details are provided in the Appendix.

By maintaining the multi-layer map, we can efficiently compute shortest transition distance between any pair of states at any time without requiring agents to execute actions in the actual environment to measure distances. Only when changes are detected in a specific layer (Geometric, Regulation, or Interference) is that layer incrementally refreshed. We only recompute distances for agent pairs within $K$ steps of the updated regions in each layer, thus avoiding expensive global updates. At computation time, the three layers are aggregated into a unified graph representation $\mathcal{G}^{(t)} = \text{Aggregate}(\mathcal{G}_{\text{geo}}^{(t)}, \mathcal{G}_{\text{reg}}^{(t)}, \mathcal{G}_{\text{int}}^{(t)})$, where $\mathcal{G}_{\text{geo}}^{(t)}$, $\mathcal{G}_{\text{reg}}^{(t)}$, and $\mathcal{G}_{\text{int}}^{(t)}$ denote the Geometric, Regulation, and Interference layer graphs at time $t$, respectively. Given the states of two agents, the shortest path can be computed by applying any shortest path algorithm on the aggregated graph $\mathcal{G}^{(t)}$. In this work, we employ Dijkstra's algorithm (Dijkstra, 1959), denoting its policy as $\pi_D$. In this regard, the distance defined in Definition 2 can be implemented as follows:

$$d_{IA}(x_1, x_2 | \mathcal{G}^{(t)}) := \sum_{t=0}^{\infty} t P(T_{x_1, x_2} = t | \pi_D, \mathcal{G}^{(t)}) \times C(T_{x_1, x_2} = t | \pi_D, \mathcal{G}^{(t)}).$$

## 4.2 Computing Cooperation Cost with an Interference Potential Field

In the course of agent cooperation, interference from other agents or environmental entities is unavoidable, undermining the stability and persistence of collaboration. Particularly in multi-agent game-theoretic scenarios, actions such as attacks or threats from adversarial agents frequently disrupt the cooperative process. Consequently, considering cooperation cost becomes crucial for selecting high-value partners for interaction.

To quantify the cooperation cost, $C(T_{s_1, s_2} = t | \pi_D)$, we require a method that can both evaluate cumulative interference along arbitrary paths in real-time and capture the threat directions and intentions of interference sources, while maintaining interpretability in the decision-making process. To this end, we propose the directional interference potential field. Unlike traditional isotropic potential fields (Khatib, 1986), our design features two key innovations: (1) directional modeling: dynamically capturing the forward threat angles of

interference sources through the effective interference distance $d_{\text{eff}}$; and (2) intent prediction: utilizing a neural network to predict attack intent vectors, enabling the potential field to adapt to dynamic adversarial behaviors. This potential field quantifies the interference strength exerted by each entity on its surrounding region, while the cooperation cost is the cumulative effect of superimposing the interference fields of all entities.For the interference potential field of a single entity $e_i$, the interference intensity at state $x$ is denoted as $I(x \mid e_i)$. We adopt a direction-aware exponential decay model (Goldsmith, 2005):

$$I(x \mid e_i) := I_{\text{base}} \, e^{- \, d_{\text{eff}} \, \lambda_{\text{base}}},$$

where the key directional modulation term is:

$$d_{\text{eff}} = d_{\text{actual}}(1 + \alpha(1 - \cos(\theta))).$$

Here, $I_{\text{base}}$ is sampled from the agent's real-time state (e.g., health and attack power). Among these parameters, $\theta$ is the angle between the neural network's predicted attack intent direction and the actual position, $\alpha$ modulates the directional influence strength, $d_{\text{actual}}$ is the Euclidean distance, and $\lambda_{\text{base}}$ is the base decay rate of the potential field's strength. When the interference source's intent is directed toward a position ($\theta \to 0$), $d_{\text{eff}}$ decreases and interference intensifies; when the intent is directed away ($\theta \to \pi$), interference weakens. This design endows the partner selection process with clear physical meaning: by avoiding paths with high directional interference to reduce cooperation cost, thereby achieving interpretable minimal-interference partner selection.Let $I(x)$ denote the total interference strength at state $x$, and $I(x|e_i)$ be the interference from entity $e_i$ (for $i \in [1, n]$) at state $x$. Then, we have:

$$I(x) = \sum_{i=1}^{n} I(x|e_i).$$

The cooperation cost is then defined as the average per-step cost along the trajectory:

$$C(T_{x_1, x_2} = t | \pi_D, \mathcal{G}^{(t)}) = \frac{1}{t} \sum_{x \in S(T_{x_1, x_2} = t | \pi_D, \mathcal{G}^{(t)})} [1 + I(x)], \quad \text{for } t > 0; \quad C(T = 0) := 0.$$

where $S(T_{x_1, x_2} = t | \pi_D)$ denotes the set of states on the t-step trajectory from $x_1$ to $x_2$ under policy $\pi_D$. In the absence of interference ($I(x) = 0$), $d_{IA}$ reduces to the expected path length $\mathbb{E}[T]$. The value of $I_{\text{base}}$ is computed based on real-time sampling of the agent's current state (e.g., health and attack power). The term $d_{\text{eff}}$ is optimized using a neural network that predicts a vector of attack intent (Casas et al., 2018). This network is trained via supervised learning to minimize the angular error between its predicted attack direction and the agent's actual trajectory. The resulting angle is then used as $\theta$ to compute the interference distance. Details are provided in the Appendix.

## 4.3 MARL with IA-KRC

Building upon the previously established multi-layer map and interference potential field, we design a dynamic grouping mechanism based on interference-aware distance. Agents are partitioned into multiple cooperative groups, where each group consists of a core leader and several followers. Within each group, we employ the QMIX value decomposition framework for training.

The process begins with leader election. To align leader selection with the communication horizon and reduce global dependence, we adopt a K-neighborhood centrality. For each agent $i$, we define the reachable neighbor count

$$N_i^{(K)} = |\{j \in \mathcal{N} : d_{IA}(x_i, x_j) \leq K\}|,$$

where $|\cdot|$ denotes the cardinality of a set. Larger $N_i^{(K)}$ indicates that agent $i$ can coordinate with more teammates within the $K$-step reachable domain, making it a better candidate for centralized communication and reducing isolated agents. We designate as leaders the top-$M$ agents with the highest $N_i^{(K)}$ scores, where $M$ is a predefined number of leaders.

Once the set of leaders is established, each non-leader agent (follower) determines its candidate leaders using the same $K$-step criterion; a leader is a candidate only if $d_{IA}(x_{\text{follower}}, x_{\text{leader}}) \le K$. To promote load balancing, the follower then affiliates with the candidate leader whose group is currently the smallest. This affiliation approach maintains balanced group sizes, mitigating resource centralization, and enhancing overall coordination efficiency.

Following group formation, we employ the QMIX value decomposition framework to train each group. Each group $g \in G$ learns a cooperative policy, where $G$ denotes the set of all cooperative groups. This is achieved by training a group-specific joint action-value function $Q_{\text{tot}}^g$ to minimize the temporal difference (TD) loss, summed over all groups:

$$\mathcal{L}(\theta) = \sum_{g \in G} \mathbb{E}_{(\boldsymbol{\tau}_g, \mathbf{a}_g, r, \boldsymbol{\tau}_g') \sim B} \left[ (y_g^{\text{tot}} - Q_{\text{tot}}^g(\boldsymbol{\tau}_g, \mathbf{a}_g; \theta))^2 \right].$$

Here, $y_g^{\text{tot}} = r + \gamma \max_{\mathbf{a}_g'} Q_{\text{tot}}^g(\boldsymbol{\tau}_g', \mathbf{a}_g'; \theta^-)$ is the TD target based on the global reward $r$, $\boldsymbol{\tau}_g$ and $\mathbf{a}_g$ are the joint history and action for group $g$, $B$ is the replay buffer, and $\theta$ and $\theta^-$ are the parameters for the online and target networks, respectively. For more details, please see the Appendix.

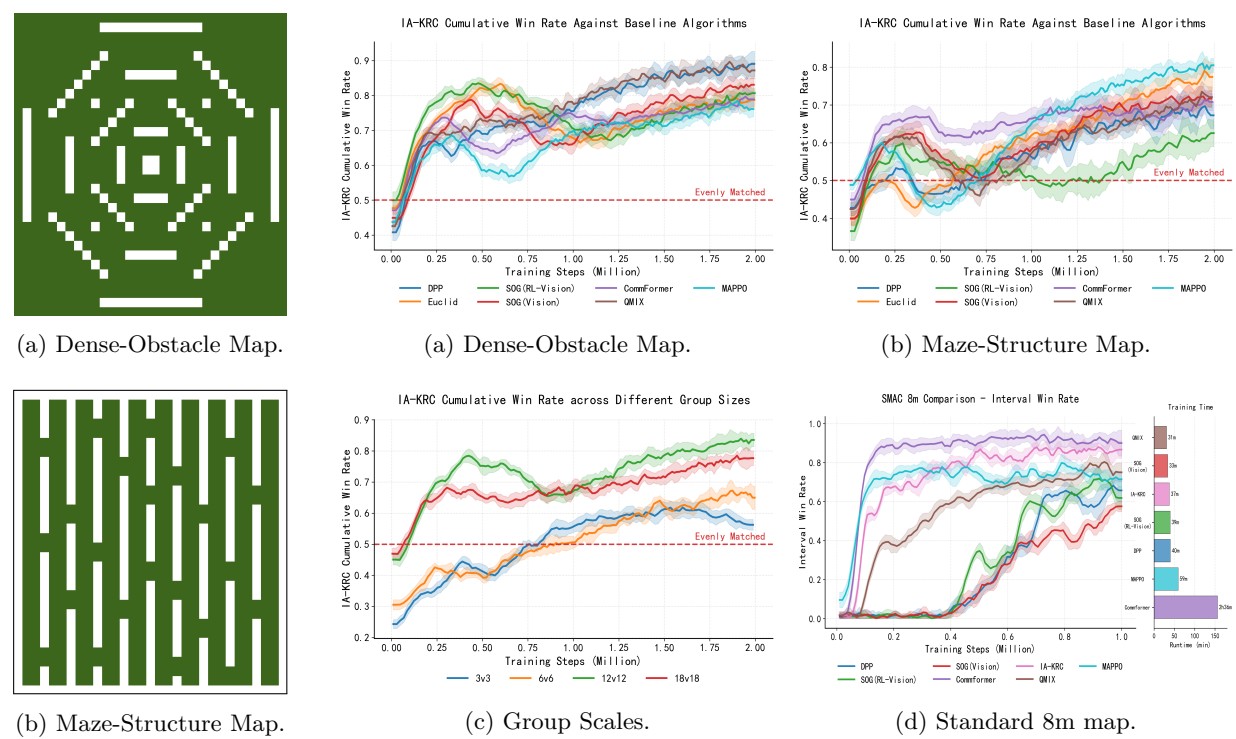

Figure 4: Custom SMACv2 maps. White regions denote obstacles.

Figure 5: Learning curves of IA-KRC and baselines over 2.0M steps. (a) Dense-Obstacle. (b) Maze-Structure. (c) Win rate across scales. (d) Standard 8m. (a)-(c): self-play; (d): built-in AI.

## 5 Experiments

We designed experiments to systematically address the following research questions about IA-KRC's effectiveness, scalability, underlying mechanisms, and generalizability:

- RQ1: (Effectiveness) Does IA-KRC outperform the main baselines in dynamic environments?

- RQ2: (Scalability and Structure) How does IA-KRC perform across different team scales and what kind of cooperative structures does it learn?

- RQ3: (Mechanism) What are the relative contributions of the two modules in IA-KRC to the overall performance?

- RQ4: (Generalization) Can IA-KRC retain its advantage in scenarios without complex topology?

To answer these questions, unlike existing methods that rely on built-in AI as evaluation baselines, we develop a self-play framework: within the same environment, two teams using different algorithms engage in adversarial play while updating their policies online. We adopt this setup because the built-in AI of the standard SMACv2 environment does not support topological occlusion (vision and attacks are not blocked by obstacles), and its closed-source nature makes rule modifications infeasible, making it difficult to validate cooperative performance under complex topologies. We provide environment and rule details in the supplementary material. The number of surviving agents at the end of each episode determines the winner. The horizon for the $K$-Step reachability domain is empirically set to $K = 9$ unless otherwise stated. All experiments are repeated 5 times with different random seeds.

### 5.1 Comparative Performance in Complex Topologies

**Experimental Setup.** To evaluate the effectiveness of IA-KRC in highly dynamic and topologically complex environments, we construct two custom SMACv2 maps (Figure 4). Figure 4 (a) depicts a symmetrical layout with dense obstacles, requiring complex path planning. Figure 4 (b) adopts a maze-like structure that creates narrow corridors and communication bottlenecks. These maps are specifically designed to amplify the challenges of decentralized coordination and test the adaptability of different communication mechanisms.

We compare IA-KRC against the following baseline methods, among which DPP, Euclid, SOG(Vision), SOG(RL-Vision), and CommFormer adopt the leader-follower framework for agent grouping:

- **DPP** Yang et al. (2020) groups agents by computing a similarity matrix to ensure diversity.

- **Euclid** Hüttenrauch et al. (2019) uses Euclidean distance between agents as the basis for grouping.

- **SOG(Vision)** Shao et al. (2022) forms groups based on mutual visibility, selects leaders randomly.

- **SOG(RL-Vision)** Shao et al. (2022) elects leaders via reinforcement learning as an extension of the SOG(Vision) method.

- **MAPPO** Yu et al. (2022) a centralized-training, decentralized-execution variant of PPO.

- **QMIX** Rashid et al. (2020) a representative value decomposition approach with a monotonic mixing network for cooperative MARL.

- **CommFormer** Hu et al. (2024) employs a graph neural network to learn communication topology end-to-end via the Gumbel-Softmax technique.

**Results and Analysis.** The learning curves in Figure 5 clearly demonstrate IA-KRC's advantages in both environments. Table 1 reports, for each method, the Dense-side and Maze-side metrics side-by-side (all metrics refer to IA-KRC): FW (final win rate), HW/S (highest win rate/step), and FL (final loss rate), all within 2.0M steps. Since draws can occur in the self-play setting, FW alone may not fully quantify algorithm effectiveness; we therefore also report FL to enable more comprehensive performance comparison.

On the Dense-Obstacle Map (Figure 5 (a)), IA-KRC achieves final win rates no lower than 77.79% (vs MAPPO), with peaks of 88.37% (vs DPP) and 88.12% (vs QMIX), demonstrating strong late-stage advantage and sustained adaptability. On the Maze-Structure Map (Figure 5 (b)), IA-KRC remains ahead under harder topology: its final win rate ranges from 62.68% (vs SOG(RL-Vision)) to 81.15% (vs MAPPO), and it maintains stable margins over Euclid, SOG(Vision), CommFormer, and QMIX.

Across both settings, we observe a recurring failure mode in baseline methods: the emergence of *isolated agents* that receive little to no valuable messages, leading to fragmented coordination. This often results in

| Method | Dense-Obstacle Map | | | Maze-Structure Map | | |
|---|---|---|---|---|---|---|
| | **FW** | **HW/S** | **FL** | **FW** | **HW/S** | **FL** |
| DPP | $88.37 \pm 3.12$ | $88.37 \pm 3.45$ / 0.87M | $2.8 \pm 1.3$ | $69.63 \pm 4.28$ | $69.63 \pm 4.10$ / 2.00M | $15.2 \pm 1.2$ |
| Euclid | $78.81 \pm 2.76$ | $83.51 \pm 2.94$ / 0.61M | $2.7 \pm 1.2$ | $77.89 \pm 3.31$ | $77.89 \pm 3.05$ / 1.99M | $6.3 \pm 1.3$ |
| SOG(Vision) | $83.51 \pm 2.67$ | $83.51 \pm 2.43$ / 1.99M | $2.6 \pm 1.1$ | $72.73 \pm 3.88$ | $72.73 \pm 3.40$ / 1.99M | $15.0 \pm 1.1$ |
| SOG(RL-Vision) | $81.11 \pm 3.01$ | $83.60 \pm 2.58$ / 0.49M | $7.5 \pm 1.4$ | $62.68 \pm 4.90$ | $62.68 \pm 4.21$ / 2.00M | $3.8 \pm 1.1$ |
| CommFormer | $79.37 \pm 2.25$ | $79.37 \pm 2.77$ / 1.99M | $7.6 \pm 1.5$ | $70.40 \pm 3.56$ | $70.40 \pm 3.12$ / 1.99M | $15.1 \pm 1.2$ |
| MAPPO | $77.79 \pm 2.64$ | $77.79 \pm 2.64$ / 2.01M | $13.2 \pm 1.6$ | $81.15 \pm 2.8$ | $81.15 \pm 2.82$ / 2.01M | $3.1 \pm 1.2$ |
| QMIX | $88.12 \pm 2.51$ | $88.12 \pm 2.51$ / 2.01M | $7.6 \pm 1.4$ | $71.98 \pm 2.6$ | $71.98 \pm 2.69$ / 2.00M | $3.1 \pm 1.1$ |

Table 1: Final cumulative win rate (FW), highest cumulative win rate (HW) within 2.0M steps for Dense-Obstacle and Maze-Structure maps. Additionally, we report final failure rate (FL) corresponding to FW.

an *avalanche effect*, where early elimination of unsupported agents cascades into total team failure. IA-KRC mitigates this by explicitly modeling both reachability and interference, dynamically selecting communication peers who are both accessible and strategically viable, thus maintaining cohesive and resilient collaboration throughout the episode.

## 5.2 Scaling and Structure Study of Multi-Agent Collaboration in IA-KRC

The two central dimensions of multi-agent collaboration are scale and group structure. Scale determines the effectiveness and efficiency of collaboration, whereas organizational structure governs how information is routed, how leader–follower roles are assigned, and whether stable, high-value coalitions can form under environmental dynamics. A mature collaboration framework should preserve a clear organizational form and stable performance at any scale—especially at large scale. Accordingly, this section examines dimensions—scale and organizational structure—and investigates how IA-KRC leverages explicit reachability and interference modeling to sustain high-quality communication and robust coordination as the collaboration setting varies in scale and organization.

**Scaling study.** To assess scalability, we conduct 2.0 M-step self-play experiments against SOG(Vision) across team sizes of 3v3, 6v6, 12v12, and 18v18. As shown in Figure 5 (c), IA-KRC achieves 65% (6v6), 86% (12v12) and 79% (18v18) by 2.0 M steps. Notably, the 3v3 scenario peaks at approximately 63% near 1.25 M steps before declining to roughly 56% by 2.0 M steps. This occurs because small team sizes enable both methods to converge toward similar near-optimal grouping structures, thereby reducing IA-KRC's relative advantage. As team size increases, the combinatorial space of group configurations expands factorially. Under this complexity, conventional methods struggle to consistently identify high-quality subgroups, whereas IA-KRC—leveraging reachability filtering and interference-aware priors—systematically constructs robust structures, thereby amplifying its performance advantage at scale.

To further understand this scalability advantage, we conducted a quantitative analysis of the computational complexity in the leader-follower framework. We measured the total computation counts for leader election and follower assignment across varying team sizes (from 4 to 64 agents) on a 64×64 map with obstacles and adversaries. As shown in Figure 6, since our algorithm only computes information within the $K$-step neighborhood for each agent, the computational complexity remains low. The total computation count grows approximately linearly with team size $N$, while the per-agent computation remains nearly constant.

**Structure study.** Beyond scale, we further examine how organizational structure affects cooperation quality. Building on the mainstream hierarchical leader–follower framework (Soni & Hu, 2018; Sheng et al., 2022), we compare alternative leader-election strategies and follower-assignment policies, and quantitatively evaluate the resulting group structures. We adopt two criteria: (1) the proportion of isolated agents, defined as the fraction of timesteps at which an agent has degree 0 in the communication graph, capturing the risk of the avalanche effect; and (2) algebraic connectivity ($\lambda_2$), the second-smallest eigenvalue of the Laplacian of each group's communication graph—larger $\lambda_2$ indicates stronger connectivity, smoother information flow, and greater robustness (Fiedler, 1973; Mohar, 1991). For completeness, the algebraic connectivity is formally

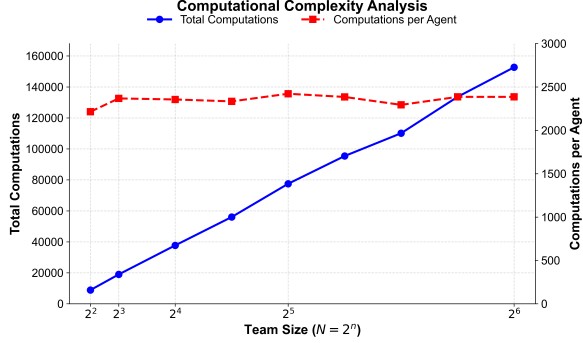

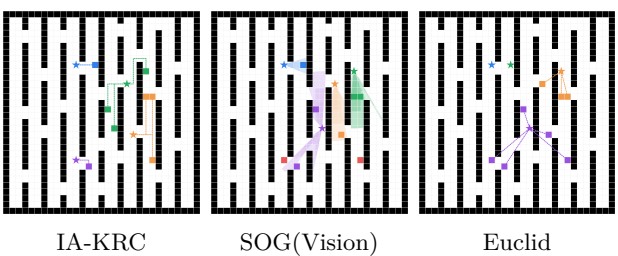

IA-KRC       SOG(Vision)       Euclid

Figure 6: Total computation counts (blue) and computations per agent (red) for leader election and follower assignment.

Figure 7: Group structure visualization: (a), (b), and (c) correspond to different algorithms; nodes with same color in same group; star=leader; red=isolated.

defined as:

$$\lambda_2(L) = \min_{x \neq 0, \, x^\top \mathbf{1} = 0} \frac{x^\top L x}{x^\top x}.$$

Here, $L = D - A$ is the unnormalized Laplacian of the communication graph, where $A$ is the adjacency matrix and $D$ is the degree matrix; $x$ is any nonzero vector with $x^\top \mathbf{1} = 0$ orthogonal to the all-ones vector $\mathbf{1}$. As shown in Table 2, IA-KRC exhibits a markedly lower isolated-agent ratio than competing methods and achieves a higher mean $\lambda_2$ with substantially lower variance, indicating reasonable grouping under complex topology that effectively mitigates the avalanche effect, while demonstrating stronger information connectivity and stable group structures that improve transmission efficiency and yield superior training outcomes. Figure 7 visualizes the grouping structures of different methods: SOG(Vision) contains many isolated agents, Euclid causes unbalanced resource allocation and inefficient grouping (e.g., cross-obstacle coordination), whereas IA-KRC strikes a better balance and discovers higher-quality structures. SOG(RL-Vision) reduces variance by learning leader selection, yet vision-based modeling limitations still yield more isolated agents and less efficient groupings. CommFormer learns group assignments via graph neural networks but struggles to quickly discover suitable structures in large, complex environments, resulting in many isolated agents and low spectral connectivity; nevertheless, its inductive bias maintains relatively low variance and comparatively stable structures. Overall, IA-KRC consistently outperforms baselines in the Maze environment;

| Method | Iso Rate | $\lambda_2$ mean | $\lambda_2$ var |
|---|---|---|---|
| IA-KRC | 0.0058 | 0.4622 | 0.0125 |
| SOG(Vision) | 0.2090 | 0.2221 | 0.2295 |
| SOG(RL-Vision) | 0.2009 | 0.2287 | 0.1242 |
| Euclid | 0.1823 | 0.2647 | 0.2198 |
| CommFormer | 0.6316 | 0.0220 | 0.0714 |

Table 2: Isolated-agent ratio (Iso Rate), mean algebraic connectivity ($\lambda_2$), and its variance (var) in Maze (12v12, 1000 groupings).

| Model Variant | $K$ | FW |
|---|---|---|
| (A) IA-KRC | 9 | $\mathbf{83.63 \pm 2.71}$ |
| (B) $K$-Step Variant | 3 | $75.48 \pm 3.02$ |
| (C) $K$-Step Variant | 6 | $79.89 \pm 2.95$ |
| (D) $K$-Step Variant | 12 | $73.66 \pm 3.87$ |
| (E) Without interference | 9 | $74.67 \pm 3.15$ |
| (F) Without $K$-Step | None | $65.53 \pm 4.76$ |

Table 3: FW of IA-KRC ablation variants on the Dense-Obstacle Map at 2.0M training steps, under self-play against Vision.

### 5.3 Ablation Study of IA-KRC Mechanisms

To assess the contributions of IA-KRC's two core components, we conducted ablation studies on the Dense-Obstacle scenario. Specifically, we vary the reachability horizon $K$ in four settings ($K = 3, 6, 9, 12$) to investigate its impact on collaboration range and prediction stability. In addition, we evaluate the effect of disabling the interference prediction module and disabling the $K$-step reachability module (replaced by

Euclidean distance) respectively, to assess its role in partner selection under dynamic disturbances and complex topologies. All ablation variants were trained in our self-play setting against the SOG(Vision) baseline; Table 3 reports the IA-KRC variants' final win rate under this setup.

**Impact of $K$-step horizon.** Table 3 shows that final performance at 2.0M steps improves as K increases from 3 (75.48%) to 6 (79.89%), peaking at K=9 (83.63%), but then decreases at K=12 (73.66%). This confirms a nonmonotonic relationship between horizon length and coordination quality. Smaller horizons such as K=3 restrict collaboration to immediate neighbors, limiting access to valued teammates farther away. This leads to lower final performance, though early results remain competitive. A moderate horizon (K=6) provides a balance of reach and stability, whereas K=9 maximizes midrange partner selection without overextending predictive uncertainty. When K increases to 12, noise accumulation in long-range cost estimates and diffuse communication graphs causes performance degradation in later stages. These results indicate that a constrained communication radius is essential for stable and effective grouping under uncertainty.

**Impact of interference prediction and $K$-Step reachability.** Model (E), which disables interference prediction while keeping K=9, reaches a final win rate of 74.67%—nearly 9 points lower than the full model. The gap widens as training progresses, as distinguishing safe from risky partners becomes critical. Without interference modeling, agents over-commit to unreliable links (e.g., enemy threats or congestion hotspots).

Model (F), which disables the $K$-Step reachability module (replacing it with a Euclidean-distance criterion), attains a final win rate of 65.53%, about 18 points lower than the full model. Because $K$-step reachability degenerates into Euclidean distance, the assessment of actual accessibility becomes overly optimistic in dense-obstacle environments; also, the Euclidean metric weakens the constraining effect of interference prediction on the true transition cost.

By contrast, the full IA-KRC—combining interference-aware partner selection with a $K$-step reachability constraint on the communication domain—effectively avoids high-cost links, yielding more cohesive and resilient communication structures and markedly stronger late-stage performance and robustness.

The ablation confirms that both the K-Step horizon and interference modeling are indispensable. Horizon length provides temporal foresight, whereas interference prediction safeguards spatial reliability. Together they enable IA-KRC to form high-utility groups, outperforming methods based on proximity or visibility.

## 5.4 Generalization to Standard Obstacle-Free Environments

To verify generalization, we evaluated IA-KRC on the obstacle-free SMACv2 8m scenario, which lacks topological constraints. We train against the built-in AI instead of using the self-play framework. All experiments were conducted on a lightweight training platform equipped with an AMD Ryzen 9 7945HX CPU, 32GB RAM, and an NVIDIA RTX 4060 GPU. We compared IA-KRC with QMIX, MAPPO, SOG, dpp, and CommFormer, with all algorithm parameters set to the default configurations provided by the authors in their open-source implementations. As shown in Figure 5 (d), even in this simplified environment where no obstacles exist and communication complexity is significantly reduced, IA-KRC maintains clear superiority, achieving faster convergence and attaining a win rate second only to CommFormer. However, under the same training conditions, CommFormer's architecture is overly "heavyweight," with training time exceeding IA-KRC by more than 4 times; in comparison, IA-KRC incurs only approximately 19% additional training time relative to other baseline algorithms. This demonstrates that IA-KRC achieves a superior balance between computational efficiency and performance.

In obstacle-free settings, the $K$-Step reachability constraint typically approximates Euclidean distance under continuous, isotropic motion, while aligning with Manhattan or Chebyshev metrics under grid-based motion primitives. Nonetheless, IA-KRC continues to outperform baseline algorithms, indicating that its advantage stems not only from geometric constraints but also from its interference-aware modeling mechanism. The dynamic influence map enables agents to effectively capture crowding, conflict zones, and temporal instability among teammates—dynamic interference factors that significantly affect collaboration even in environments without physical obstacles. In contrast, the SOG and DPP method exhibits prolonged exploration phases, limiting improvements in coordination efficiency. CommFormer demonstrates rapid growth and strong performance but suffers from excessive resource consumption, restricting its practical applicability. MAPPO

and QMIX perform reasonably in the 8m setting but overall remain inferior to IA-KRC. These results fully confirm that IA-KRC's core mechanisms—reachability filtering and interference prediction—generalize effectively beyond complex topologies, providing robust and significant performance advantages even in environments with minimal structural constraints.

## 6 Conclusion

We proposed IA-KRC, a hierarchical communication mechanism for multi-agent reinforcement learning that integrates $K$-Step reachability and interference-aware modeling to form effective, low-conflict collaboration structures. Central to our approach is a multi-layer map framework that decouples slow-changing environmental topology from rapidly evolving agent adversarial dynamics, ensuring real-time computation of interference-aware $K$-step reachable distances to enable effective cooperation.

Extensive experiments across diverse SMACv2 scenarios validate the effectiveness and generality of IA-KRC. In complex environments with Dense-Obstacle and Maze-Structure settings, IA-KRC consistently outperformed baselines such as CommFormer and MAPPO, converging faster and achieving higher win rates. Even in the standard 8m scenario without obstacles, IA-KRC maintained an advantage over Euclidean baselines, highlighting its robust generalization beyond structural complexity. These results demonstrate that IA-KRC is not only effective under spatial constraints but also excels in capturing behavioral interference in open environments.

## Acknowledgments

This work was supported by the Qiyuan Innovation Program under Grant S20230201023.

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

## APPENDIX

This appendix provides detailed specifications for the key components of the Interference-Aware $K$-step Reachable Communication (IA-KRC) framework, including its core algorithm, the neural network for interference prediction, and the hyperparameters used in the experiments.

### A.1   Implementation Details of the IA-KRC Algorithm

**Key Variables and Functions.**   Before presenting the algorithm, we define the key functions and variables used:

**Variables:**

- $d_{IA}(s_1, s_2)$: Interference-aware shortest transition distance (Definition 2)

- $\mathcal{S}_{IA}(s_1, K)$: Interference-aware $K$-step reachable region (Definition 3)

- $N_i^{(K)}$: Reachable neighbor count within $K$-step domain for agent $i$

- $\mathcal{L}$: Set of elected leaders

- $\mathcal{F}$: Set of followers (non-leader agents)

**Functions:**

- `update_from_sight`(): Updates geometry layer with visual observations

- `update`() on $L_r$: Updates regulation layer using agent transitions

- `confidence_refresh`($\cdot; L_g, L_r$): Confidence-driven asynchronous updater over layers

- `interference_prediction_module`(): Produces interference costs for $L_i$

- `aggregate_graph`():   Aggregates geometry obstacles, regulation constraints, and interference weights; sets obstacle $\Rightarrow \infty$, free base $= 1$, and non-obstacle edges $w(u, v) \leftarrow 1 + f_{\text{influence}}(M, v)$ when interference is enabled

- `dijkstra_reachable`(): Shortest-path query on aggregated graph to get $\mathcal{S}_{IA}(s, K)$

The complete IA-KRC algorithm is presented in Algorithm 1.

---

**Algorithm 1** IA-KRC Algorithm

---

1: **Input:** Agent states $\{s_i\}_{i=1}^N$, enemy observations $E_{obs}$, environment observations $env_{obs}$, parameters $K, M$
2: **Output:** Collaborative groups $G = \{G_1, G_2, ..., G_M\}$
3: {Phase 1: Multi-Layer Map Update}
4: Initialize layers: $L_g$, $L_r$, $L_i$
5: Extract `agent_pos`, `map_matrix` from observations
6: $L_g$.`update_from_sight`(`agent_pos`, `map_matrix`, `sight_range`)
7: $L_r$.`update`(`agent_transitions`)
8: `confidence_refresh`(`experience_stats`; $L_g, L_r$)
9: {Phase 2: Interference Prediction}
10: **if** interference enabled **then**
11:    `interference_costs` $\leftarrow$ `interference_prediction_module`($E_{obs}$, `agent_pos`)
12:    $L_i$.`update`(`interference_costs`)
13: **end if**
14: {Phase 3: Graph Aggregation and Reachability}
15: $\mathcal{G}_{agg} \leftarrow$ `aggregate_graph`($L_g, L_r, L_i$)
16: **for** each agent $i$ **do**
17:    $\mathcal{S}_{IA}(s_i, K) \leftarrow$ `dijkstra_reachable`($\mathcal{G}_{agg}$, $s_i$, $K$)
18:    $N_i^{(K)} \leftarrow |\{\, j \neq i : s_j \in \mathcal{S}_{IA}(s_i, K) \,\}|$
19: **end for**
20: {Phase 4: Leader Election and Follower Assignment}
21: $\mathcal{L} \leftarrow$ top $M$ agents with highest $N_i^{(K)}$ scores
22: Initialize groups: $G_l \leftarrow \{l\}$ for each $l \in \mathcal{L}$
23: $\mathcal{F} \leftarrow \{1, ..., N\} \setminus \mathcal{L}$
24: **for** each follower $f \in \mathcal{F}$ **do**
25:    $\mathcal{L}_{cand} \leftarrow \{l \in \mathcal{L} : s_f \in \mathcal{S}_{IA}(s_l, K)\}$
26:    **if** $\mathcal{L}_{cand} \neq \emptyset$ **then**
27:      $l^* \leftarrow \arg\min_{l \in \mathcal{L}_{cand}} |G_l|$
28:      Add $f$ to group $G_{l^*}$
29:    **end if**
30: **end for**
31: **return** $G = \{G_1, G_2, ..., G_M\}$

---

## A.2 Multi-Layer Map Module

**Geometry Layer.** Maintains physical connectivity on a grid using agents' visual observations and line-of-sight within a circular perception radius. Newly observed obstacles and free cells update only local neighborhoods; a full grid rebuild is triggered only when novel obstacles are discovered. This layer exposes `update_from_sight` to incrementally refresh obstacle cells, visited cells, and adjacency.

---

**Algorithm 2** Geometry layer: LOS-based incremental update

---

1: Input: agent position $p$, map matrix $M$, sight range $R$
2: **for** each cell $q$ with $\|q - p\| \leq R$ and LOS($p, q$) **do**
3:    **if** $M[q] = -1$ **then**
4:      mark $q$ as obstacle; update local adjacency
5:    **else**
6:      mark $q$ as visited free cell; update local adjacency
7:    **end if**
8: **end for**
9: If new obstacles found: rebuild local grid connections; otherwise keep incremental updates

---

**Regulation Layer.** Stores environment rule-based connectivity inferred from actual agent transitions (e.g., doors, traffic rules). At each step and for each agent, it logs the tuple $(s_t, a_t, s_{t+1}, \texttt{success})$ derived from the chosen action and observed position change. If $\texttt{success}$ is true, it updates a directed edge $s_t \to s_{t+1}$ in the regulation graph; failures are forwarded to the confidence-driven asynchronous mechanism for localized revalidation without directly modifying raw geometry.

---

**Algorithm 3** Regulation layer: transition logging with success/failure

---

1: Input: time $t$, agent id $i$, current state $s_t$, action $a_t$, next state $s_{t+1}$, flag success
2: Append $(i, t, s_t, a_t, s_{t+1}, success)$ to transition_log
3: **if** success **then**
4:     **if** edge $(s_t, s_{t+1})$ not in _*adj* **then**
5:         add directed edge $s_t \to s_{t+1}$ with base weight $w(s_t, s_{t+1}) \leftarrow 1$
6:     **else**
7:         optionally update running statistics for $(s_t, s_{t+1})$
8:     **end if**
9:     send record to confidence mechanism: update_stats(edge=$(s_t, s_{t+1})$, succ$\leftarrow$1, total$\leftarrow$1)
10: **else**
11:     send record to confidence mechanism: update_stats(edge=$(s_t, s_{t+1})$, succ$\leftarrow$0, total$\leftarrow$1)
12: **end if**

---

**Confidence-Driven Asynchronous Mechanism.** Tracks per-edge reliability statistics (success/total), maintains a FIFO of recently blocked candidates, and triggers localized refresh based on a confidence threshold $\tau_c$ and decay factor $\eta_{\text{upd}}$. When reliability for edge $(u, v)$ falls below the threshold (e.g., persistent failures), it marks $(u, v)$ as a blocked candidate and schedules a future revalidation with decayed frequency; otherwise the edge remains usable. This mechanism operates externally to the three layers (Geometry, Regulation, Interference) as an asynchronous updater, supplying obstacle candidates to aggregation without touching raw geometry.

---

**Algorithm 4** Confidence-driven asynchronous update: reliability-based blocking with FIFO

---

1: Input: stats map $S[(u, v)] = (succ, total)$, threshold $\tau_c$, decay $\eta_{\text{upd}}$, FIFO queue $Q$
2: **for** each edge $(u, v)$ in $S$ **do**
3:     $r \leftarrow \frac{succ}{\max(1, total)}$
4:     **if** $r < \tau_c$ **then**
5:         mark $(u, v)$ as blocked; push $(u, v)$ into FIFO $Q$ (evict oldest if capacity reached)
6:         schedule revalidation time $t_{\text{next}} \propto \eta_{\text{upd}}$
7:     **else**
8:         keep $(u, v)$ active (unblocked)
9:     **end if**
10: **end for**

---

**Graph Aggregation and Query.** Aggregation combines: (i) geometry-layer obstacles, (ii) regulation-layer constraints and blocked candidates proposed by the confidence mechanism (treated as obstacles), and (iii) interference weights from Section A.3. The base edge weight policy is: obstacle/non-transferable $\Rightarrow \infty$; otherwise base $= 1$. With interference enabled, the influence map $M$ (see Section A.3) assigns costs to all non-obstacle transitions by starting from 1 and then computing $w(u, v) \leftarrow 1 + f\_\text{influence}(M, v)$ per the formula, yielding dynamic traversal costs. Consequently, the aggregated result satisfies: free space has base 1, obstacles are $\infty$, and influenced regions are data-dependent.

**Algorithm 5** Aggregate-and-Query under policy $\pi_D$ (geometry + regulation + interference)

---

1: Build obstacle set $\mathcal{O} \leftarrow$ geometry_obstacles $\cup$ regulation_constraints $\cup$ confidence_blocked_candidates

2: **for** each candidate edge $(u, v)$ **do**
3:     **if** $(u, v) \in \mathcal{O}$ **then**
4:         $w(u, v) \leftarrow \infty$
5:     **else**
6:         $w(u, v) \leftarrow 1$
7:     **end if**
8: **end for**
9: **if** interference enabled **then**
10:     obtain influence map $M$ from Section A.3; for non-obstacle $(u, v)$ set $w(u, v) \leftarrow 1 + f\_\text{influence}(M, v)$ (cf. A.3)
11: **end if**
12: Run Dijkstra with weights $w$ to obtain costs and $\mathcal{S}_{IA}(s, K)$
13: Return reachable set and costs

---

### A.3   Interference Prediction Module

The Interference Prediction Module quantifies the traversal risk by modeling the influence of adversarial agents as a potential field. In this field, high-threat enemies generate high-cost regions. The module's core is the calculation of a potential field for each enemy, which incorporates directional influence via a predicted attack intent angle $\theta$. This influence is modulated by a threat level, which is heuristically determined from the enemy's current state and recent actions. As detailed in Algorithm 6, a neural network predicts an attack intent vector for each enemy to derive the angle $\theta$. The resulting path cost map informs the IA-KRC framework's reachability calculations and agent coordination. Figure 8 shows an example of the dynamically computed transition cost map.

### Key Variables and Functions for Algorithm 2
**Variables:**

- $E_{obs}$: Visual observations of enemies.

- $M_{influence}$: A grid representing the cumulative enemy influence across the map.

- $M_{cost}$: A grid representing the pathfinding cost for each location.

- $e$: An individual enemy entity.

- $I_{\text{base}}$: Dynamically computed base influence strength for an enemy.

- $\lambda_{\text{base}}$: Influence decay rate (hyperparameter).

- $\alpha$: Angle influence factor (hyperparameter).

- $\theta_e$: Predicted attack intent angle for enemy $e$.

- $d_{\text{eff}}(p_1, p_2, \theta_e)$: Effective distance considering the predicted attack angle.

- $d_{\text{actual}}(p_1, p_2)$: Euclidean distance between two points.

- *cost_multiplier*: A factor to scale influence into path cost (hyperparameter).

**Functions:**

- `extract_enemies`($E_{obs}$): Parses observations to get a list of enemy entities and their states.

- `calculate_influence`($e$): Computes the dynamic base influence strength $I_{\text{base}}$ of an enemy based on its attributes (e.g., health, recent actions).

- `predict_attack_intent`($e$): Predicts the attack intent angle $\theta_e$ for an enemy using a neural network.

- `normalize_map`($M$): Normalizes map values to a standard range.

---

**Algorithm 6** Interference Prediction and Cost Calculation

---

1: **Input:** Enemy observations $E_{obs}$, hyperparameters $\alpha, \lambda_{\text{base}}$
2: **Output:** Path cost map $M_{cost}$
3: {Phase 1: Initialize Maps}
4: Initialize $M_{influence}$ with zeros.
5: Initialize $M_{cost}$ with ones.
6: {Phase 2: Calculate Enemy Influence}
7: $Enemies \leftarrow \texttt{extract\_enemies}(E_{obs})$
8: **for** each enemy $e$ in $Enemies$ **do**
9:     $I_{\text{base}} \leftarrow \texttt{calculate\_influence}(e)$ {Compute influence from state}
10:     $\theta_e \leftarrow \texttt{predict\_attack\_intent}(e)$ {Predict directional intent}
11:     **for** each cell $p$ on the map within influence range of $e$ **do**
12:         $d_{\text{actual}} \leftarrow d_{\text{actual}}(p, e.position)$
13:         $d_{\text{eff}} \leftarrow d_{\text{actual}} \times (1 + \alpha(1 - \cos(\theta_e)))$ {Calculate effective distance}
14:         $I(p|e) \leftarrow I_{\text{base}} \times \exp(-\lambda_{\text{base}} \times d_{\text{eff}})$
15:         $M_{influence}[p] \leftarrow M_{influence}[p] + I(p|e)$
16:     **end for**
17: **end for**
18: {Phase 3: Compute Path Cost Map}
19: $M_{cost} \leftarrow 1.0 + cost\_multiplier \times M_{influence}$ {Convert influence to traversal cost}
20: For each obstacle position $p_{obs}$: $M_{cost}[p_{obs}] \leftarrow \infty$
21: $M_{cost} \leftarrow \texttt{normalize\_map}(M_{cost})$
22: **return** $M_{cost}$

---

**Neural Network and Parameter Details**

This section provides further implementation details for the key components of the Interference Prediction Module.

**Neural Network for Attack Intent Prediction.** The attack intent angle $\theta_e$ for each enemy is derived from a predicted attack vector, which is generated by a dedicated neural network. This network takes an enemy's state as input and outputs a 2D vector representing its most likely direction of attack. The network architecture is a feed-forward Multi-Layer Perceptron (MLP) with two hidden layers:

- **Input Layer:** A flattened vector representing the enemy's state, including its recent trajectory (last 10 positions) and current health.
- **Hidden Layers:** The first hidden layer consists of 128 neurons with a ReLU activation function, followed by a second hidden layer of 64 neurons, also with ReLU activation.
- **Output Layer:** A linear layer with 2 neurons, producing the (x, y) components of the predicted attack intent vector.

**Training Process.** The attack intent prediction network is trained via supervised learning on data collected during simulation. For each enemy at each time step, we create a training sample consisting of its current state (the network input) and its actual movement vector over the next time step (the ground truth). The network is trained to minimize the angular difference between its predicted attack vector $\mathbf{v}_{\text{pred}}$ and the

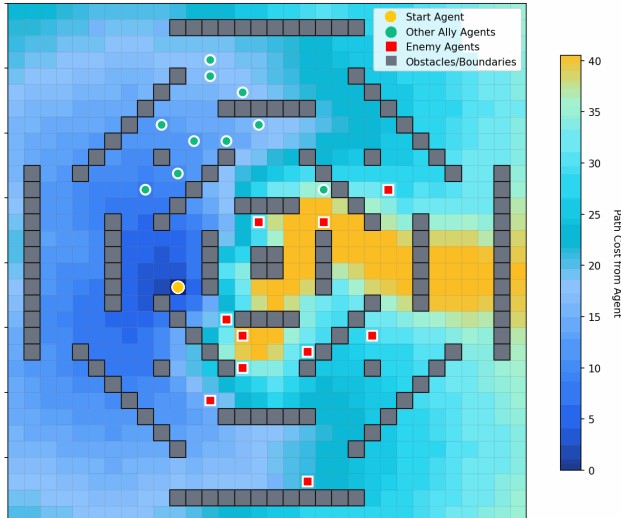

Figure 8: Transition cost map computed from the start agent (yellow) after multi-layer map and interference map processing. Blue regions indicate low traversal cost due to high ally density and minimal enemy interference, while yellow regions represent high traversal cost caused by distant locations and strong adversarial agent interference.

ground truth movement vector $\mathbf{v}_{\mathrm{true}}$. The loss function is defined as the negative cosine similarity:

$$\mathcal{L}_{\mathrm{intent}} = 1 - \frac{\mathbf{v}_{\mathrm{pred}} \cdot \mathbf{v}_{\mathrm{true}}}{\|\mathbf{v}_{\mathrm{pred}}\| \|\mathbf{v}_{\mathrm{true}}\|}$$

Training is performed using the Adam optimizer with a learning rate consistent with the main policy training (see Table 4).

**Dynamic Influence Calculation.** The base influence strength $I_{\mathrm{base}}$ for each enemy is computed dynamically via the `calculate_influence`(e) function in Algorithm 2:

$$I_{\mathrm{base}} = I_{\mathrm{config}} \times T_e$$

where $I_{\mathrm{config}}$ is the configured base influence strength (default 2.0), and $T_e$ is the threat level: $T_e = \frac{T_{\mathrm{move}} + T_{\mathrm{attack}} + T_{\mathrm{health}} + T_{\mathrm{mobility}}}{4}$, with $T_{\mathrm{move}} = \min(1.0, |\mathrm{trajectory}|/50.0)$, $T_{\mathrm{attack}} = \min(1.0, |\mathrm{attacks}|/10.0)$, $T_{\mathrm{health}} = \mathrm{normalized\_health}$, and $T_{\mathrm{mobility}} = \min(1.0, \bar{d}_{\mathrm{recent}}/3.0)$.

**Effective Distance ($d_{\mathrm{eff}}$).** The effective distance $d_{\mathrm{eff}} = d_{\mathrm{actual}}(1 + \alpha(1 - \cos(\theta_e)))$ creates a forward-facing cone of influence, where $\theta_e$ is the angle between the predicted attack intent vector and the vector from enemy position to location $p$. The hyperparameter $\alpha$ controls the directional effect strength.

## A.4 IA-KRC Training Details in Multi-Agent Reinforcement Learning

The IA-KRC framework is integrated within a multi-agent reinforcement learning (MARL) system that employs value decomposition methods for cooperative policy learning. This section details the training architecture and implementation specifics of how IA-KRC's dynamic grouping mechanism is incorporated into the MARL training pipeline, extending traditional value-based methods to accommodate interference-aware coordination.

**Group-Specific Value Function.** The core of our framework is the monotonic decomposition of the group's joint action-value function. $Q^g_{\mathrm{tot}}$ is represented as a monotonic combination of the individual utility

functions $Q_i$ for all agents $i \in g$. This property is crucial for decentralized execution, as it guarantees that a greedy action selection by each agent based on its local $Q_i$ corresponds to the maximization of the joint $Q_{\text{tot}}^g$. The relationship is formalized as $Q_{\text{tot}}^g = f_g(\{Q_i\}_{i \in g}, s)$, where $f_g$ is a mixing network specific to group $g$ and conditioned on the global state $s$. This architecture implies that each group effectively learns its own cooperative policy, tailored to its members and the current state, while still contributing to the global team objective.

**Agent Architecture.** Each agent utilizes a recurrent neural network (RNN) with an attention mechanism. This network processes the agent's local observation history $\tau_i$ to maintain a hidden state $h_i^t$. At each timestep, the RNN's input includes the agent's current observation, its previous action, and any messages received from its group leader. The resulting hidden state $h_i^t$ is then fed into a feed-forward layer to compute the per-action Q-values $Q_i(\tau_i, \cdot)$.

**Mixing Network.** The framework employs a mixing network architecture, specified as `flex_qmix` in the configuration, to combine individual $Q_i$ values into the joint $Q_{\text{tot}}^g$. Consistent with QMIX, the weights of this mixing network are generated by a hypernetwork that takes the global state $s$ as input, allowing the mixing function to adapt to different environmental conditions. To enhance training stability, the mixing weights are normalized using a softmax function.

**Training Process.** The system is trained end-to-end by minimizing the total TD loss, summed across all dynamically formed groups, as defined in the main paper. Experiences are collected using parallel runners and stored in a replay buffer. The learner (`msg_q_learner`) samples mini-batches to perform updates. The TD target for each group $g$ is computed as:

$$y_g^{\text{tot}} = r + \gamma \max_{\mathbf{a}_g'} Q_{\text{tot}}^g(\boldsymbol{\tau}_g', \mathbf{a}_g'; \theta^-)$$

where $\theta$ and $\theta^-$ are the parameters of the online and target networks, respectively. The target network is periodically updated with the online network's parameters every 200 episodes.

## A.5  Hyperparameter Settings

Table 4: Key Hyperparameter Settings

| Parameter | Value | Parameter | Value | Parameter | Value |
|---|---|---|---|---|---|
| **General RL Parameters** | | | | | |
| Learning Rate | 5e-4 | Optimizer | Adam | Discount Factor ($\gamma$) | 0.99 |
| Batch Size | 32 | Replay Buffer Size | 5000 | Target Update Interval | 200 ep. |
| Epsilon Start | 1.0 | Epsilon Finish | 0.05 | Epsilon Anneal Time | 500k steps |
| Experience Buffer Size | 10000 | Opponent Learning Rate | 1e-3 | | |
| **IA-KRC Framework and Interference Prediction** | | | | | |
| K-step Horizon ($K$) | 9 | Number of Leaders ($N_L$) | 3 | Cost Multiplier | 1.5 |
| Interference Decay ($\lambda_{\text{base}}$) | 0.3 | Angle Influence ($\alpha$) | 0.5 | Influence Range | 5.0 |
| Base Influence Strength | 2.0 | Intent Net Hidden Dim 1 | 128 | Intent Net Hidden Dim 2 | 64 |
| **Agent and Mixer Architecture** | | | | | |
| RNN Hidden Dim | 64 | Attention Heads | 4 | Hypernet Embed Dim | 128 |
| Mixing Net Dim | 32 | | | | |

## A.6  Avalanche Effect Demonstration

This section demonstrates the avalanche effect through a competitive evaluation between IA-KRC and Vision algorithms after 2.0M training steps. The red team represents Vision agents, while the green team represents IA-KRC agents. The following four stages illustrate how initial tactical advantages can cascade into decisive victory through the avalanche effect mechanism.

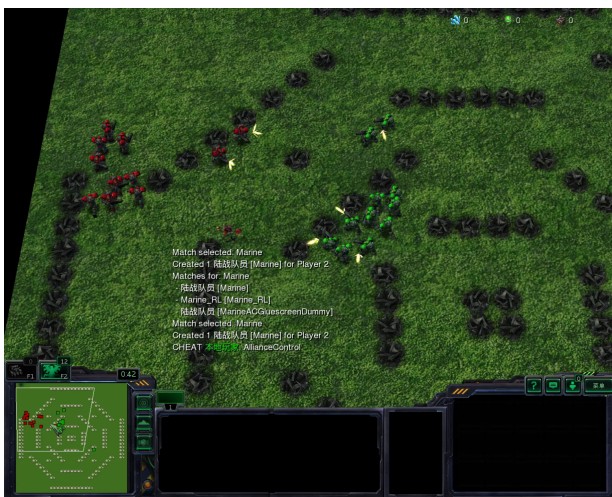

Figure 9: Stage 1: Due to vision limitations, the red team (Vision) forms two isolated agents. These isolated agents cannot communicate or coordinate with other teammates, resulting in low collaboration efficiency and becoming surrounded by the green team (IA-KRC).

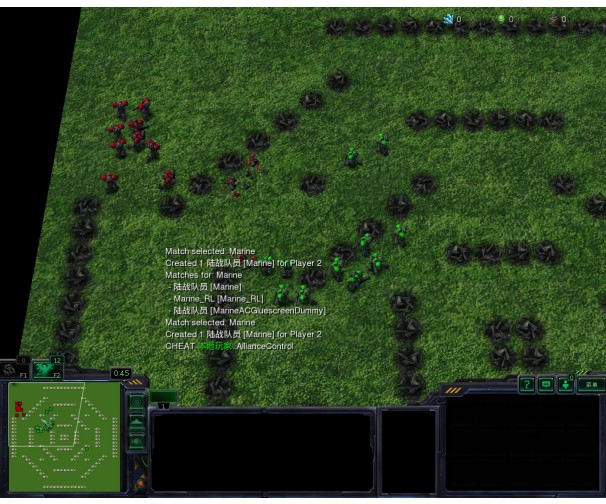

Figure 10: Stage 2: The isolated Vision agents are eliminated through concentrated fire from IA-KRC (IA-KRC loses 1 agent while Vision loses 3 agents), creating an 11v9 situation. The avalanche effect begins to manifest.

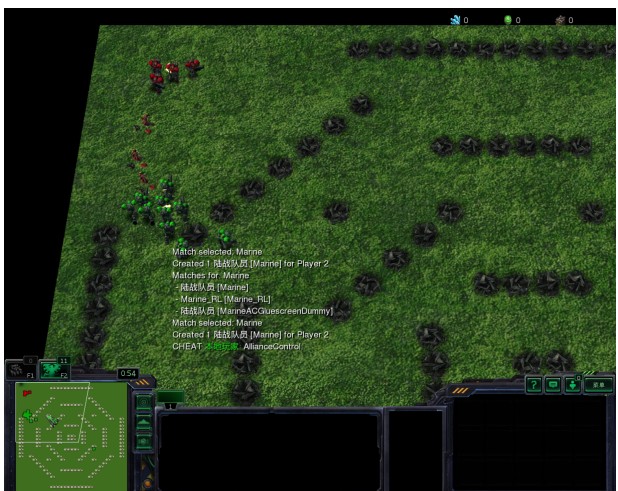

Figure 11: Stage 3: Leveraging the numerical advantage, IA-KRC adopts an aggressive strategy. With superior firepower, they continuously expand the combat capability gap.

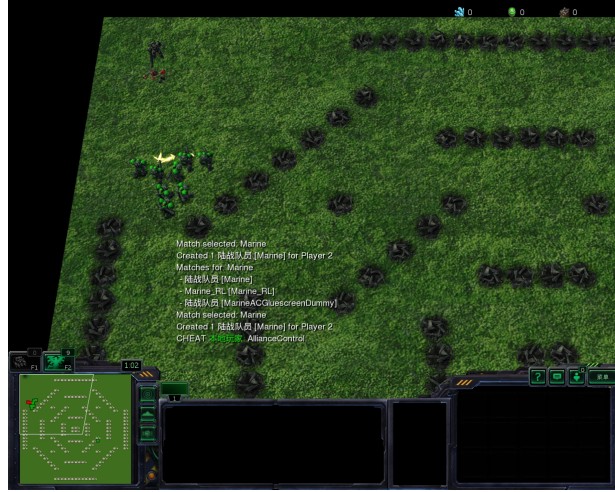

Figure 12: Stage 4: IA-KRC achieves complete victory, eliminating all Vision agents at the cost of only 4 agents, demonstrating the effectiveness of interference-aware coordination in multi-agent combat scenarios.

