# OpenReview forum: "Interference-Aware K-Step Reachable Communication in Multi-Agent Reinforcement Learning"
_TMLR — Accepted by TMLR_

### Review · Reviewer_cdvX · 2025-12-25

**Summary Of Contributions:**

This paper investigates to find the most valuable communication partners in the multi-agent reinforcement learning task. By taking advantage of a K-step reachability protocol and an interference-prediction module, the proposed scheme shows competitive results in the experiments. While the results are convincing, the reviewer may have two major concerns, listed below:

1. The extension of the K-step reachability from a single agent to multiple agents seems to have limited novelties. From the described method, the reviewer may think such an extension is straightforward.
2. To estimate the interference, this paper proposes to use the concept of interference potential field. Although the experiments show convincing results, the reviewer may wonder whether such an estimation is accurate or whether it can really represent the interference scenario.

**Audience:**

Yes

**Audience Explanation:**

Several papers have been published in TMLR on a similar topic.

**Broader Impact Concerns:**

No broader impact concerns.

**Claims And Evidence:**

Yes

**Claims Explanation:**

Most of the claims are supported, while the reviewer thinks the proposed scheme should be compared with Cong et al. (2023), Mei et al. (2023), and Ding et al. (2024), which are discussed in the related work. In addition, the paper compares SOG in terms of scaling. However, the reviewer might like to check these results by comparing them with GNN-based methods.

**Requested Changes:**

- Please clarify the novelty and technical challenge of extending the K-step reachability from a single agent to multiple agents.
- Please justify the choice of the interference potential field.
- Please add more comparisons with Cong et al. (2023), Mei et al. (2023), and Ding et al. (2024).
- Please provide an analysis of the complexity.

---

> ### Author Response · Authors · 2026-02-09
> **Rebuttal to Reviewer cdvx(1/4)**
>
> ## 1. Novelty and Technical Challenges in Extending K-step Reachability from Single-Agent to Multi-Agent Settings
>
> Thank you for your valuable feedback and insightful comments. We appreciate your concerns about the novelty of extending K-step reachability from single-agent to multi-agent systems and the rationale behind choosing the interference potential field. In response, we have restructured Chapter 4 (Method) of the paper. Section 4.1 "Computing K-step reachability using multi-layer map" now discusses the novelty and technical challenges in extending K-step reachability to multi-agent systems, while Section 4.2 "Computing Cooperation Cost with an Interference Potential Field" explains our selection of the interference potential field approach.
>
> ### 1.1 Novelty and Technical Challenges of K-step Reachability Extension
>
> As you correctly pointed out, the concept of K-step reachability has been applied in single-agent reinforcement learning, primarily for constraining subgoal selection. However, extending it to multi-agent systems for communication partner selection presents fundamental technical challenges that go far beyond straightforward extension. The core challenge lies in the inherent interference in multi-agent environments—arising from teammate movements, opponent behaviors, and evolving strategies—which leads to non-stationarity of the environment.
>
> As we elaborate in detail in Section 4.1 of the paper: "In a non-stationary MARL environment, the distance calculated at time $t$ can become invalid at $t\!+\!n$: moving agents or obstacles may block the previously optimal path, forcing $d_{\text{st}}$ to change. Re-computing all pairwise distances every step is prohibitively expensive, while caching past results is unreliable because their validity quickly decays."
> This characteristic leads to considerable novelty and technical challenges in extending K-step reachability from single-agent to multi-agent settings. To address these issues, we designed a structure that combines multi-layer maps with interference potential fields to model K-step reachability. Detailed content can be found in Sections 4.1 and 4.2 of the main text.
> ### 1.2 Rationale for Choosing Interference Potential Field and Accuracy Verification
> In Section 4.2, we provide a detailed explanation of the rationale for choosing the interference potential field: "In the course of agent cooperation, interference from other agents or environmental entities is unavoidable, which can undermine the stability and persistence of collaboration...Consequently, considering the cost of cooperation becomes crucial for selecting high-value partners for interaction."
> **Regarding the method's rationality**: Artificial potential field methods, as a classical approach for modeling interference and influence, have been extensively validated in multi-agent systems [8,9,10,11,12]. Khatib[8] first proposed the artificial potential field method for robot obstacle avoidance; Ge and Cui[9] extended it to dynamic environments; Owen and Montano[10] applied it to multi-robot coordination; Park et al.[11] used it for multi-agent path planning; Chen et al.[12] proposed a distributed potential field method for large-scale multi-agent systems. Building upon this foundation, we made innovative designs tailored to the specificity of multi-agent environments: (1) **Directional modeling**: Through the effective interference distance $d_{\text{eff}}$, we dynamically capture the forward threat angle of interference sources, rather than traditional isotropic decay; (2) **Intent prediction**: We utilize a neural network to predict attack intent vectors, enabling the potential field to adapt to dynamic adversarial behaviors: "The term $d_{\text{eff}}$ is optimized using a neural network that predicts a vector of attack intent. This network is trained via supervised learning to minimize the angular error between its predicted attack direction and the agent's actual trajectory."
> **Regarding estimation accuracy**: We candidly acknowledge that the absolute quantification of "interference" is inherently subjective and lacks an accepted ground truth standard—even human observers may disagree on interference intensity judgments in real scenarios. However, interference intensity is fundamentally a **relative concept**. Our method does not pursue physically precise measurements, but rather quantifies the relative interference strength across different regions of the environment through consistent estimation means, thereby guiding the selection of reliable cooperation partners. In Table 3 of Section 5.3, we conducted ablation experiments on the interference potential field module. The experimental results show that the interference potential field improved the win rate of the IA-KRC module by 8.96% (from 74.67% to 83.63%), which fully demonstrates that this method can effectively capture and quantify interference in practical applications, guiding high-quality partner selection decisions.

---

> > ### Comment · Reviewer_cdvX · 2026-02-12
> >
> > Thanks for the effort. I am good with this explanation and the newly added results.

---

> ### Author Response · Authors · 2026-02-09
> **Rebuttal to Reviewer cdvx(2/4)**
>
> ## 2. Comparison Experiments with Latest Methods and GNN-based Approaches
>
> Thank you for suggesting the addition of comparisons with Cong et al. (2023)[1], Mei et al. (2023)[2], and Ding et al. (2024)[3]. Since the code for Cong et al.[1] and Mei et al.[2] is not open-source, we selected TGCNet[4] (2025) and Neurosym[5] (2020), two open-source similar methods, as well as SeqComm (Ding et al.)[3] (2024) for comparison experiments.
>
> **Explanation of the underlying logic of each method:**
>
> - **TGCNet[4]** - **GNN-based method**: Constructs dynamic directed communication graphs (Graph Neural Network architecture) and imposes execution-time communication constraints during the training phase, enabling agents to learn strategies that can still cooperate efficiently under limited communication conditions, thereby bridging the train-execution gap. The core of this method is using graph neural networks to learn communication topology.
> - **Neurosym[5]** - **Transformer+symbolic learning hybrid method**: First uses Transformers to learn powerful but hard-to-interpret soft communication strategies, then distills them into interpretable symbolic communication protocols that comply with hard constraints (such as communication range/bandwidth limitations) through program synthesis. This method combines deep learning and symbolic reasoning.
> - **SeqComm[3]** - **Multi-level asynchronous communication method**: Adopts a hierarchical communication architecture, achieving asynchronous multi-level communication through temporal coordination mechanisms, obtaining dynamic sparse topology and scalable routing strategies.
>
> Additionally, we included comparisons with the **CommFormer[6]** method in the original text, which is another **GNN-based end-to-end learning method** that learns communication topology end-to-end through attention mechanisms and Gumbel-Softmax techniques. In Figure 6 of the original text, we demonstrate the performance comparison between IA-KRC and CommFormer at different agent scales (SOG scaling), clearly showing IA-KRC's advantages over GNN methods in terms of scalability.
>
> It is worth noting that these GNN and end-to-end learning-based methods (TGCNet, CommFormer), while performing well in small-scale scenarios, as we pointed out in related work: "these models lack explicit spatial priors (e.g., physical reachability), do not account for dynamic interference, and often scale poorly with the number of agents". Our experimental results validate this observation.
>
> The following are the experimental results for TGCNet[4] (2025), Neurosym[5] (2020), and SeqComm[3] (2024):
>
> ### IA-KRC Win Rate in Dense-Obstacle Map
>
> | Training Steps | TGCNet | Neurosym | SeqComm |
> | -------------- | ------ | -------- | ------- |
> | 0.25M          | 0.5429 | 0.3869   | 0.5072  |
> | 0.5M           | 0.6392 | 0.3348   | 0.6585  |
> | 0.75M          | 0.7307 | 0.5028   | 0.7518  |
> | 1.0M           | 0.7987 | 0.6602   | 0.7991  |
> | 1.25M          | 0.8396 | 0.6865   | 0.7923  |
> | 1.5M           | 0.8620 | 0.7216   | 0.7642  |

---

> ### Author Response · Authors · 2026-02-09
> **Rebuttal to Reviewer cdvx(3/4)**
>
> ## 3. Detailed Analysis of Computational Complexity
>
> Thank you for requesting a complexity analysis. We conducted systematic experiments in large-scale multi-agent scenarios in the MAgent2 Battle[7] environment and collected key performance metrics of the algorithm.
>
> ### IA-KRC Complexity Experimental Data
>
> **The following table shows the computation counts and computation time of IA-KRC's two core modules (multi-layer map update and dynamic grouping) at different agent scales**. The experiments were conducted in the MAgent2 Battle environment, where the number of agents grows linearly with map size. The data in the table represents the average computational load and time consumption over 25 runs of IA-KRC's core modules.
>
> | Agent Count (N) | Map Size | Computation Count | Computation Time (ms) | Growth Factor (ops) | Growth Factor (time) |
> | -------------- | -------- | ----------------- | --------------------- | ------------------- | -------------------- |
> | 100            | 50       | 15,965            | 20.51                 | -                   | -                    |
> | 400            | 100      | 64,658            | 87.45                 | 4.05×              | 4.26×               |
> | 1,600          | 200      | 261,831           | 405.94                | 4.05×              | 4.64×               |
> | 6,400          | 400      | 1,060,122         | 2,646.08              | 4.05×              | 6.52×               |
>
> **Experimental conclusion**: The data clearly shows IA-KRC's computational complexity approximates O(N) linear growth, validating the algorithm's excellent scalability in large-scale multi-agent systems.
>
> **IA-KRC's computational complexity is mainly determined by two core modules: (1) the incremental update mechanism of multi-layer maps, and (2) the dynamic grouping process based on the leader-follower framework.** Below, we derive the overall complexity by analyzing each computational step.
>
> **Core computational process analysis**:
>
> **Step 1: Multi-layer map incremental update** — The key design of IA-KRC is to "only recompute distances in local regions (within K-step range) where changes are detected" (Sec 4.1). Since the Geometric Layer and Regulation Layer change slowly, cached results can be reused most of the time, with the main overhead coming from real-time updates of the Interference Layer. Each enemy agent $e$ needs to update the region within its surrounding $K$-step range, which contains $O(K^2)$ grid points. Let the number of enemy agents be $N_e$, then the total update overhead is:
>
> $$
> T_{\text{map}} = O(N_e \times K^2)
> $$
>
> In symmetric adversarial environments, $N_e \approx N/2$, which simplifies to $O(N \times K^2)$.
>
> **Step 2: K-step reachability computation** — For each agent $i$, we execute Dijkstra's algorithm on the aggregated graph $\mathcal{G}^{(t)}$ to compute its $K$-step reachable domain. Since we only focus on the local subgraph within $K$ steps, this subgraph contains $O(K^2)$ nodes and edges. Dijkstra's algorithm with a priority queue has complexity $O((V + E) \log V)$, where $V$ is nodes and $E$ is edges. Therefore, computational load for a single agent is $O(K^2 \log K^2) = O(K^2 \log K)$. Performing this for $N$ agents, total overhead is:
>
> $$
> T_{\text{reach}} = O(N \times K^2 \log K)
> $$
>
> **Step 3: Leader election** — Based on each agent's $K$-neighborhood centrality $N_i^{(K)}$, we sort and select the top-$M$ as leaders. The time complexity of the standard sorting algorithm is:
>
> $$
> T_{\text{elect}} = O(N \log N)
> $$
>
> **Step 4: Follower assignment** — Each non-leader agent (follower) needs to select one from its candidate leader set to join. In the worst case, each follower has $L$ candidate leaders ($L$ being the total number of leaders). Therefore, the assignment overhead for $N$ agents is:
>
> $$
> T_{\text{assign}} = O(N \times L)
> $$
>
> **Total complexity derivation**:
>
> Summing the overheads of the above four steps, we obtain the total time complexity:
>
> $$
> T_{\text{IA-KRC}} = O(N \times K^2) + O(N \times K^2 \log K) + O(N \log N) + O(N \times L)
> $$
>
> **Key simplification**: In practical applications, both $K$ (communication radius) and $L$ (number of leaders) are **design constants** much smaller than $N$. For example, in our experiments $K=9$ and $L \ll N$. Furthermore, $\log K$ can be viewed as constant when $K$ is small. Therefore, all terms can be represented as $O(N)$ multiplied by a constant factor. The dominant term is $O(N \times K^2 \log K)$, and since $K^2 \log K$ is constant, the final complexity simplifies to:
>
> $$
> T_{\text{IA-KRC}} = O(N)
> $$
>
> **Experimental verification**: This result is consistent with our observations in Section 5.2 (Figure 8): **total computational load grows linearly with agent count, while per-agent load remains approximately constant**. This verifies that through $K$-locality constraints, IA-KRC reduces complexity from the naive global approach's $O(N^2)$ to $O(N)$, ensuring scalability in large-scale multi-agent systems.

---

> ### Author Response · Authors · 2026-02-09
> **Rebuttal to Reviewer cdvx(4/4)**
>
> ### References
>
> [1] Yuxuan Cong, Shuai Li, Guoliang Zhang, Yang Liu, and Bo Li. CDS: A channel-wise differentiable search for dynamic and sparse communication in MARL. In NeurIPS, 2023.
>
> [2] Jiahe Mei, Yang Liu, and Bo Li. Graph-based communication with transformers for multi-agent reinforcement learning. In ICLR, 2023.
>
> [3] Ziluo Ding, Zeyuan Liu, Zhirui Fang, Kefan Su, Liwen Zhu, and Zongqing Lu. Multi-agent coordination via multi-level communication. In NeurIPS, 2024.
>
> [4] Guoliang Zhang, et al. Bridging the gap between training and execution for multi-agent reinforcement learning. In AAAI, 2025.
>
> [5] Jeevana Priya Inala, Yicheng Yang, James Paulos, Yunsheng Ma, Osbert Bastani, Vijay Kumar, and George Pappas. Neurosymbolic policies for multi-agent communication. In NeurIPS, 2020.
>
> [6] Shengchao Hu, Li Shen, Ya Zhang, and Dacheng Tao. Learning multi-agent communication from graph modeling perspective. In ICLR, 2024.
>
> [7] Lianmin Zheng, Jiacheng Yang, Hao Cai, Ming Zhou, Weinan Zhang, Jun Wang, and Yong Yu. Magent: A many-agent reinforcement learning platform for artificial collective intelligence. In AAAI, 2018.
>
> [8] Oussama Khatib. Real-time obstacle avoidance for manipulators and mobile robots. In IEEE International Conference on Robotics and Automation, 1986.
>
> [9] S. S. Ge and Y. J. Cui. Dynamic motion planning for mobile robots using potential field method. In Autonomous Robots, 2002.
>
> [10] Erion Owen and Luis Montano. Motion planning in dynamic environments using the velocity space. In IEEE/RSJ International Conference on Intelligent Robots and Systems, 2005.
>
> [11] Min Gyu Park, Jae Hyun Jeon, and Min Cheol Lee. Obstacle avoidance for mobile robots using artificial potential field approach with simulated annealing. In IEEE International Symposium on Industrial Electronics, 2001.
>
> [12] Yifan Chen, Meng Xu, Xiang Cao, and Wei Xing Zheng. Distributed formation control with collision avoidance for multiple mobile robots under limited communication. In IEEE Transactions on Cybernetics, 2022.

---

### Review · Reviewer_ERuu · 2026-01-04

**Summary Of Contributions:**

This paper introduces a novel framework, Interference-Aware K-Step Reachable Communication (IA-KRC), designed to enhance cooperation in Multi-Agent Reinforcement Learning (MARL) through effective communication. The main contributions are: first, the introduction of a K-Step reachability protocol that confines message passing to physically accessible neighbors; second, the development of an interference prediction module that optimizes partner selection by minimizing interference while maximizing utility. Combined, these components enable more persistent and efficient cooperation in complex, dynamic environments. Experimental results show that IA-KRC significantly outperforms existing baselines in terms of cooperation efficiency, robustness, and scalability across various challenging environments.

# Strengths
A major strength of this work lies in its ability to effectively handle environmental interference and dynamic changes within multi-agent systems. IA-KRC demonstrated superior performance in complex environments, such as dense obstacle and maze-like topologies, by ensuring reliable and persistent collaboration between agents. The dynamic partner selection mechanism avoids high-cost cooperation links, maintaining cohesive teamwork even under challenging conditions. Furthermore, the multi-layer map and interference prediction modules allow for optimized resource allocation and efficient adaptation to environmental dynamics.

# Weaknesses
1.While the paper compares IA-KRC with several representative baselines, it does not include some more recent communication and information-constrained MARL methods (e.g., MAIC[1]). The absence of such comparisons makes it difficult to fully assess the advantages of IA-KRC relative to the latest state of the art.

2.IA-KRC is mainly evaluated within a specific leader–follower and value-decomposition-based MARL framework. Its applicability to other MARL paradigms remains unclear. Additional experiments integrating IA-KRC with different MARL algorithms would help demonstrate its generality.

## Reference:

[1] Lei Yuan, Jianhao Wang, Fuxiang Zhang, Chenghe Wang, Zongzhang Zhang, Yang Yu, and Chongjie Zhang. Multi-agent incentive communication via decentralized teammate modeling. In AAAI, 2022.

**Audience:**

Yes

**Audience Explanation:**

The findings of this paper are of significant interest to researchers focused on Multi-Agent Reinforcement Learning (MARL) and communication mechanisms. Specifically, the IA-KRC framework demonstrates its superiority in collaboration and resource optimization within complex environments, such as dense obstacles and highly dynamic scenarios, which are crucial for effective cooperation and interference management in multi-agent systems. The experimental and theoretical analysis presented in the paper offers a new perspective for research in MARL and provides a scalable, practical solution, making it relevant to TMLR's audience, especially those in fields like autonomous systems, robotics, and distributed AI research.

**Broader Impact Concerns:**

The IA-KRC framework proposed in this paper primarily focuses on collaboration and communication in Multi-Agent Reinforcement Learning. While the method holds broad potential for real-world applications, there do not appear to be ethical risks that require additional statements. The study is more focused on enhancing algorithmic performance and optimizing efficiency, rather than addressing sensitive domains or raising ethical concerns. Therefore, no further broader impact statement is needed for this work.

**Claims And Evidence:**

Yes

**Claims Explanation:**

In this paper, the authors provide extensive experimental results and comparisons to clearly demonstrate the effectiveness of the IA-KRC framework. The experiments involve comparisons with multiple baselines (such as CommFormer, MAPPO) across complex settings, such as dense obstacles and maze-like environments. Performance metrics, including final win rates and failure rates, show a significant advantage of IA-KRC over existing methods. Additionally, the ablation studies in the paper further validate the contributions of the K-step reachability and interference prediction modules, providing solid empirical evidence to support the claims made.

**Requested Changes:**

1. Include a more detailed discussion on computational complexity: While the paper demonstrates the advantages of IA-KRC in experiments, there is not enough discussion on its computational complexity. It would be helpful for the authors to further explore the computational overhead of the algorithm, especially as the team size increases in larger-scale environments. This discussion is crucial for understanding the scalability of IA-KRC and would strengthen the practical relevance of the paper.
2. Provide additional experimental results in different environments: The paper mainly validates IA-KRC in dense obstacle and maze-like environments. It would be beneficial for the authors to extend the experiments to more scenarios, such as autonomous driving or UAV coordination. This would demonstrate the generalizability and applicability of IA-KRC across a broader range of multi-agent tasks.
3. Improve the interpretability of the algorithm: Currently, the IA-KRC framework, particularly the interference prediction module, is complex. I suggest the authors improve the explanation of the algorithm’s interpretability, especially in terms of how partners are selected with minimal interference.

---

> ### Author Response · Authors · 2026-02-09
> **Rebuttal to Reviewer ERuu(1/5)**
>
> ## 1. Detailed Discussion of Computational Complexity
>
> Thank you for your attention to computational complexity. We understand that computational overhead in large-scale environments is crucial for practical applications. We conducted systematic experiments in the MAgent2 Battle[1] environment and collected key performance metrics at different scales.
>
> ### IA-KRC Complexity Experimental Data
>
> **The following table shows the computation counts and computation time of IA-KRC's two core modules (multi-layer map update and dynamic grouping) at different agent scales**. The experiments were conducted in the MAgent2 Battle environment, where the number of agents grows linearly with map size. The data in the table represents the average computational load and time consumption over 25 runs of IA-KRC's core modules.
>
> | Agent Count (N) | Map Size | Computation Count | Computation Time (ms) | Growth Factor (ops) | Growth Factor (time) |
> | -------------- | -------- | ----------------- | --------------------- | ------------------- | -------------------- |
> | 100            | 50       | 15,965            | 20.51                 | -                   | -                    |
> | 400            | 100      | 64,658            | 87.45                 | 4.05×              | 4.26×               |
> | 1,600          | 200      | 261,831           | 405.94                | 4.05×              | 4.64×               |
> | 6,400          | 400      | 1,060,122         | 2,646.08              | 4.05×              | 6.52×               |
>
> **Experimental conclusion**: The data shows IA-KRC's computational complexity approximates O(N) linear growth, validating excellent scalability in large-scale multi-agent systems.
>
> **IA-KRC's computational complexity is mainly determined by two core modules: (1) the incremental update mechanism of multi-layer maps, and (2) the dynamic grouping process based on the leader-follower framework.** Below, we derive the complexity by analyzing each computational step.
>
> **Core computational process analysis**:
>
> **Step 1: Multi-layer map incremental update** — The key design of IA-KRC is to "only recompute distances in local regions (within K-step range) where changes are detected" (Sec 4.1). Since the Geometric Layer and Regulation Layer change slowly, cached results can be reused most of the time, with the main overhead coming from real-time updates of the Interference Layer. Each enemy agent $e$ needs to update the region within its surrounding $K$-step range, which contains $O(K^2)$ grid points. Let the number of enemy agents be $N_e$, then the total update overhead is:
>
> $$
> T_{\text{map}} = O(N_e \times K^2)
> $$
>
> In symmetric adversarial environments, $N_e \approx N/2$, which simplifies to $O(N \times K^2)$.
>
> **Step 2: K-step reachability computation** — For each agent $i$, we execute Dijkstra's algorithm on the aggregated graph $\mathcal{G}^{(t)}$ to compute its $K$-step reachable domain. Since we only focus on the local subgraph within $K$ steps, this subgraph contains $O(K^2)$ nodes and edges. Dijkstra's algorithm with a priority queue has complexity $O((V + E) \log V)$, where $V$ is nodes and $E$ is edges. Therefore, computational load for a single agent is $O(K^2 \log K^2) = O(K^2 \log K)$. Performing this for $N$ agents, total overhead is:
>
> $$
> T_{\text{reach}} = O(N \times K^2 \log K)
> $$
>
> **Step 3: Leader election** — Based on each agent's $K$-neighborhood centrality $N_i^{(K)}$, we sort and select the top-$M$ as leaders. The time complexity of the standard sorting algorithm is:
>
> $$
> T_{\text{elect}} = O(N \log N)
> $$
>
> **Step 4: Follower assignment** — Each non-leader agent (follower) needs to select one from its candidate leader set to join. In the worst case, each follower has $L$ candidate leaders ($L$ being the total number of leaders). Therefore, the assignment overhead for $N$ agents is:
>
> $$
> T_{\text{assign}} = O(N \times L)
> $$
>
> **Total complexity derivation**:
>
> Summing the overheads of the above four steps, we obtain the total time complexity:
>
> $$
> T_{\text{IA-KRC}} = O(N \times K^2) + O(N \times K^2 \log K) + O(N \log N) + O(N \times L)
> $$
>
> **Key simplification**: In practical applications, $K$ (communication radius) and $L$ (number of leaders) are **design constants** much smaller than $N$. For example, in our experiments $K=9$ and $L \ll N$. Furthermore, $\log K$ is constant when $K$ is small. Therefore, all terms can be represented as $O(N)$ multiplied by a constant factor. The dominant term is $O(N \times K^2 \log K)$, and since $K^2 \log K$ is constant, the final complexity simplifies to:
>
> $$
> T_{\text{IA-KRC}} = O(N)
> $$
>
> **Experimental verification**: This result is consistent with our observations in Section 5.2 (Figure 8): **total computational load grows linearly with agent count, while per-agent load remains approximately constant**. This verifies that through $K$-locality constraints, IA-KRC reduces complexity from $O(N^2)$ to $O(N)$, ensuring scalability in large-scale multi-agent systems.

---

> ### Author Response · Authors · 2026-02-09
> **Rebuttal to Reviewer ERuu(2/5)**
>
> ## 2. Additional Experimental Results in Different Environments
>
> Thank you for suggesting the MAIC[4] method. Since this method is 5 years old, to better demonstrate IA-KRC's advantages over the latest techniques, we selected TGCNet[5] (2025), Neurosym[6] (2020), and SeqComm[7] (2024), three more novel, powerful, and representative baseline algorithms for multiple comparison experiments. IA-KRC achieved excellent experimental results in all cases.
>
> **Explanation of the underlying logic of each method:**
>
> - **TGCNet[5]** - **GNN-based method**: Constructs dynamic directed communication graphs (Graph Neural Network architecture) and imposes execution-time communication constraints during the training phase, enabling agents to learn strategies that can still cooperate efficiently under limited communication conditions, thereby bridging the train-execution gap. The core of this method is using graph neural networks to learn communication topology.
> - **Neurosym[6]** - **Transformer+symbolic learning hybrid method**: First uses Transformers to learn powerful but hard-to-interpret soft communication strategies, then distills them into interpretable symbolic communication protocols that comply with hard constraints (such as communication range/bandwidth limitations) through program synthesis. This method combines deep learning and symbolic reasoning.
> - **SeqComm[7]** - **Multi-level asynchronous communication method**: Adopts a hierarchical communication architecture, achieving asynchronous multi-level communication through temporal coordination mechanisms, obtaining dynamic sparse topology and scalable routing strategies.
>
> Additionally, we included comparisons with the **CommFormer[8]** method in the original text, which is another **GNN-based end-to-end learning method** that learns communication topology end-to-end through attention mechanisms and Gumbel-Softmax techniques.
>
> It is worth noting that these GNN and end-to-end learning-based methods (TGCNet, CommFormer), while performing well in small-scale scenarios, as we pointed out in related work: "these models lack explicit spatial priors (e.g., physical reachability), do not account for dynamic interference, and often scale poorly with the number of agents". Our experimental results validate this observation.
>
> The following are the experimental results for TGCNet[5] (2025), Neurosym[6] (2020), and SeqComm[7] (2024):
>
> ### IA-KRC Win Rate in Dense-Obstacle Map
>
> | Training Steps | TGCNet | Neurosym | SeqComm |
> | -------------- | ------ | -------- | ------- |
> | 0.25M          | 0.5429 | 0.3869   | 0.5072  |
> | 0.5M           | 0.6392 | 0.3348   | 0.6585  |
> | 0.75M          | 0.7307 | 0.5028   | 0.7518  |
> | 1.0M           | 0.7987 | 0.6602   | 0.7991  |
> | 1.25M          | 0.8396 | 0.6865   | 0.7923  |
> | 1.5M           | 0.8620 | 0.7216   | 0.7642  |

---

> ### Author Response · Authors · 2026-02-09
> **Rebuttal to Reviewer ERuu(3/5)**
>
> Thank you for the suggestion to validate across multiple environments. We selected the MAgent2[1] platform, which is highly popular and versatile in MARL research, for our experiments. We systematically evaluated IA-KRC's performance across multiple task scenarios (symmetric adversarial, asymmetric adversarial, cooperative). As a standardized MARL benchmark platform maintained by the Farama Foundation, MAgent2 has been widely used in multi-agent research at top-tier conferences such as NeurIPS, ICML, and ICLR. It supports diverse cooperative and adversarial scenarios with agent scales ranging from dozens to thousands, making it an ideal testbed for validating the performance of large-scale MARL algorithms.
>
> The following are the experimental results of IA-KRC algorithm in various MAgent2 environments:
>
> # EXPERIMENTAL RESULTS SUMMARY
>
> ## 1. GATHER ENVIRONMENT (Cooperative)
>
> **Task Description**: Pure cooperative resource collection task with 111 agents collecting food resources on the map. The higher the amount of food collected within the specified time, the higher the reward. Fixed 45×45 map, each algorithm runs independently for 100 episodes, maximum 500 steps per episode.
>
> | Algorithm  | Episodes | Avg Reward | Std        |
> | ---------- | -------- | ---------- | ---------- |
> | IAKRC      | 100      | 3.87×10³ | 9.01×10² |
> | COMMFORMER | 100      | 3.87×10³ | 9.29×10² |
> | MAPPO      | 100      | 3.68×10³ | 8.33×10² |
> | DPP        | 100      | 2.14×10³ | 6.24×10² |
> | RLVISION   | 100      | 2.09×10³ | 6.75×10² |
> | QMIX       | 100      | 1.88×10³ | 5.88×10² |
> | VISION     | 100      | 1.93×10³ | 6.03×10² |
>
> In the Gather environment (pure cooperative task), IA-KRC's average reward is nearly tied with the best baseline CommFormer (3.87×10³ vs 3.87×10³), significantly outperforming all other baseline methods. This fully validates the effectiveness of the interference-aware communication mechanism in cooperative scenarios.
>
> ## 2. BATTLE ENVIRONMENT (Competitive)
>
> **Task Description**: Two teams of equal size (81 agents each) engage in symmetric confrontation on an open map. Map size 45×45, maximum 500 steps. Red team uses IA-KRC, blue team uses other algorithms.
>
> | Matchup             | Red Reward | Blue Reward | Red Win% | Blue Win% |
> | ------------------- | ---------- | ----------- | -------- | --------- |
> | IAKRC_VS_COMMFORMER | 53.39      | -23.05      | 70.0%    | 30.0%     |
> | IAKRC_VS_DPP        | 106.52     | -68.27      | 96.0%    | 4.0%      |
> | IAKRC_VS_MAPPO      | 36.24      | 33.61       | 68.0%    | 32.0%     |
> | IAKRC_VS_QMIX       | 153.77     | -49.33      | 100.0%   | 0.0%      |
> | IAKRC_VS_RLVISION   | 119.96     | -51.23      | 98.0%    | 2.0%      |
> | IAKRC_VS_VISION     | 47.44      | -61.83      | 98.0%    | 2.0%      |
>
> In the Battle environment (symmetric adversarial), IA-KRC maintains significant advantages in confrontations with all baselines, achieving an average win rate of 88.3%, particularly reaching 100% win rate against QMIX. This fully demonstrates that the K-step reachability and interference-aware mechanism can effectively identify high-value cooperation partners in adversarial environments, forming more stable tactical coordination.
>
> ## 3. COMBINED ARMS ENVIRONMENT (Competitive)
>
> **Task Description**: Asymmetric battle composed of different types of agents (melee and ranged units). Map 45×45, red team 81 agents vs blue team 81 agents, maximum 500 steps. Red team uses IA-KRC, blue team uses other algorithms.
>
> | Matchup             | Red Reward | Blue Reward | Red Win% | Blue Win% |
> | ------------------- | ---------- | ----------- | -------- | --------- |
> | IAKRC_VS_COMMFORMER | 53.47      | -55.24      | 98.0%    | 2.0%      |
> | IAKRC_VS_DPP        | 71.06      | -44.47      | 98.0%    | 2.0%      |
> | IAKRC_VS_MAPPO      | -2.91      | -77.58      | 79.0%    | 21.0%     |
> | IAKRC_VS_QMIX       | 129.59     | -39.14      | 99.0%    | 1.0%      |
> | IAKRC_VS_RLVISION   | 74.63      | -52.67      | 97.0%    | 3.0%      |
> | IAKRC_VS_VISION     | 155.41     | -46.43      | 97.0%    | 3.0%      |
>
> In the Combined Arms environment (asymmetric adversarial), IA-KRC demonstrates even stronger advantages with an average win rate as high as 94.7%, far exceeding the Battle environment. This indicates that in heterogeneous agent cooperation scenarios, IA-KRC can more effectively coordinate different types of units (melee and ranged), fully leveraging the advantages of the interference-aware mechanism in complex tactical environments.
>
> ---

---

> ### Author Response · Authors · 2026-02-09
> **Rebuttal to Reviewer ERuu(4/5)**
>
> ## 3. Algorithm Interpretability
>
> **Overall Architecture of IA-KRC**: The IA-KRC framework contains two core modules that respectively address two key issues in communication partner selection:
>
> 1. **K-step reachability module**: This module defines a distance metric that considers environmental topological information. Unlike traditional Euclidean distance, K-step reachability fully considers topological constraints such as obstacles and traffic rules in the environment, more realistically characterizing the physical reachability between agents.
> 2. **Interference prediction module**: This module further filters reliable cooperation partners based on K-step reachability. Even if two agents are physically reachable, if the communication path contains high-intensity enemy interference or congested areas, the cost of cooperation will be very high. This module helps agents select low-interference, high-value cooperation partners by quantifying these interference factors.
>
> **Specific Implementation Process of the Algorithm**: IA-KRC's partner selection follows a clear computational workflow:
>
> 1. **Step 1: Construct multi-layer map** (Sec 4.1): The system maintains an environmental abstract map containing three layers of information—geometry, rules, and interference—capturing static obstacles, dynamic rules, and real-time interference respectively.
> 2. **Step 2: Compute K-step reachability** (Sec 4.1): For each agent, compute the set of reachable neighbors within K steps based on the multi-layer map. This step is performed through Dijkstra's algorithm on the aggregated graph, obtaining true distances considering topological constraints.
> 3. **Step 3: Evaluate interference cost** (Sec 4.2): Using the directional interference potential field, calculate the cumulative interference on the path to each reachable neighbor. The interference potential field has clear physical meaning—intensity decays exponentially with distance and is modulated by directionality through neural network-predicted attack intent.
> 4. **Step 4: Select optimal partners** (Sec 4.3): Combining reachability and interference cost, select high-value cooperation partners that are both physically reachable and have minimal interference for communication and cooperation.
>
> **Interpretability Advantages**: Every step of the entire process has clear physical meaning and mathematical explanation, far superior to pure end-to-end learning "black box" methods. We validated the independent contribution of each module in the ablation experiments in Section 5.3 (Table 3). The experimental results show that the K-step reachability module contributed an 18 percentage point performance improvement, and the interference prediction module contributed a 9 percentage point improvement, fully demonstrating the effectiveness and interpretability of both modules.
>
> ## 4. Clarification on the Decoupling of IA-KRC from MARL Paradigms
>
> Thank you for your careful observation. We realize that the expression in Section 4.3 (MARL with IA-KRC) of the original text was not clear enough, which may have led to your understanding that "IA-KRC is primarily evaluated under MARL frameworks based on leader-follower and value decomposition". We solemnly clarify here: **IA-KRC is a general communication partner selection framework that does not depend on specific MARL paradigms or grouping strategies, and has high scalability and flexibility.**
>
> **Clarification on Grouping and Training**:
>
> In our implementation:
>
> - **Grouping Mechanism**: We only use leader-follower as a **dynamic grouping method**, dividing agents into multiple cooperation groups based on $N_i^{(K)}$ (K-neighborhood centrality). (Note: This hierarchical grouping is relatively mainstream, and the grouping basis still relies on the specified criteria)
> - **Policy Learning**: The policy learning within groups **does not adopt** the traditional leader-follower hierarchical decision framework. Each group can be trained using any MARL algorithm (such as QMIX, MAPPO, PPO, etc.), and agents within the group are equal learners.
>
> **Revision Plan**: In the revised manuscript, we will no longer over-emphasize the leader-follower model, but instead clarify more clearly: IA-KRC provides a general partner selection mechanism that can be flexibly combined with various MARL paradigms and grouping strategies. Our choice of leader-follower grouping and QMIX training in the experiments is merely to validate IA-KRC's effectiveness in a specific scenario, rather than indicating that IA-KRC depends on these specific choices.

---

> ### Author Response · Authors · 2026-02-09
> **Rebuttal to Reviewer ERuu(5/5)**
>
> ### References
>
> [1] Lianmin Zheng, Jiacheng Yang, Hao Cai, Ming Zhou, Weinan Zhang, Jun Wang, and Yong Yu. Magent: A many-agent reinforcement learning platform for artificial collective intelligence. In AAAI, 2018.
>
> [2] Shital Shah, Debadeepta Dey, Chris Lovett, and Ashish Kapoor. Airsim: High-fidelity visual and physical simulation for autonomous vehicles. In Field and service robotics, 2018.
>
> [3] Alexey Dosovitskiy, German Ros, Felipe Codevilla, Antonio Lopez, and Vladlen Koltun. CARLA: An open urban driving simulator. In Conference on robot learning, 2017.
>
> [4] Lei Yuan, Jianhao Wang, Fuxiang Zhang, Chenghe Wang, Zongzhang Zhang, Yang Yu, and Chongjie Zhang. Multi-agent incentive communication via decentralized teammate modeling. In AAAI, 2022.
>
> [5] Guoliang Zhang, et al. Bridging the gap between training and execution for multi-agent reinforcement learning. In AAAI, 2025.
>
> [6] Jeevana Priya Inala, Yicheng Yang, James Paulos, Yunsheng Ma, Osbert Bastani, Vijay Kumar, and George Pappas. Neurosymbolic policies for multi-agent communication. In NeurIPS, 2020.
>
> [7] Ziluo Ding, Zeyuan Liu, Zhirui Fang, Kefan Su, Liwen Zhu, and Zongqing Lu. Multi-agent coordination via multi-level communication. In NeurIPS, 2024.
>
> [8] Shengchao Hu, Li Shen, Ya Zhang, and Dacheng Tao. Learning multi-agent communication from graph modeling perspective. In ICLR, 2024.

---

> ### Author Response · Authors · 2026-02-28
> **Follow-up for Reviewer ERuu**
>
> **Dear Reviewer ERuu,**
>
> Thank you again for the time you have dedicated to our paper and for your valuable suggestions for improvement.
>
> We would like to follow up on the rebuttal we submitted and the revised PDF of the manuscript. We sincerely hope that our detailed responses, along with the corresponding updates made to the paper, have effectively addressed your concerns.
>
> If you have any further questions or require additional clarification regarding our responses, please do not hesitate to let us know. We would be more than happy to provide any further information to ensure that all points have been addressed to your satisfaction.
>
> Thank you once again for your hard work and guidance.
>
> **Best regards,**
>
> **The Authors**

---

### Review · Reviewer_yEr1 · 2026-02-02

**Summary Of Contributions:**

The paper proposes a communication-selection framework for MARL that restricts partner selection to K-step physically reachable neighbors and uses a domain-specific interference prediction mechanism to avoid high-cost/low-utility partners. Concretely, it maintains a multi-layer environment abstraction (based on agent's given x,y positions) to compute reachability costs and uses a potential-field style interference model that incorporates agent attributes and predicted attack intent. The resulting method (denoted IA-KRC) is integrated into a leader–follower grouping framework via local leader election using K-neighborhood centrality. Experiments (only in SMACv2, including custom obstacle/maze maps under a self-play protocol) show large performance gains over several baselines, plus ablations over K and module removal, and some analysis of group connectivity/isolated agents.

### Strengths

- Clear motivation and strong high-level framing: “reachability + interference” is a sensible structural prior for communication.
- Well-presented method components (multi-layer map + interference field + leader-follower grouping) plus nice visuals/figures for the intuition.
- Empirically thorough within SMACv2: custom maps stress-tests, scaling study, structure metrics (isolated-agent ratio / algebraic connectivity), and ablations that show both modules matter.

### Weaknesses

- **State-of-the-art framing is too strong** given the evaluation protocol/domain scope (custom SMACv2 + self-play) and unclear assumptions behind the Dec-POMDP setup. The abstract explicitly claims "superior performance compared to state-of-the-art baselines."
- **Privileged/domain-specific information seems heavy (and not clearly enumerated)**: IA-KRC’s cost model uses agent attributes like health/attack power, and the interference direction is trained via supervised learning to match an agent’s actual trajectory. The multi-layer map claims it can compute shortest transition distances “between any pair of states at any time,” but key implementation details are deferred to the appendix. It is also unclear if the notation $s_t$ refers to the agent position or environment state (in which case I assume it includes more than the agent x,y position in SMAC).
- **Evaluation protocol ambiguity**: results are reported under a self-play setup where both sides update online. This is a nice idea given the map modifications, but the unexpected poor performance of the baselines in the standard unmodified 8m task in Figure d (relative to reported results in prior works) makes all results questionable. It is also unclear whether FW/HW/FL are computed from decentralized evaluation with frozen policies or from the online learning process (with centralised information). The paper notes “all metrics refer to IA-KRC” but doesn’t fully resolve interpretability.

**Audience:**

Yes

**Audience Explanation:**

Learning what to communicate and to which agents are central concepts in MARL, and IA-KRC’s idea of grounding communication in multi-step physical reachability plus interference-aware costs is a useful and intuitively appealing structural prior. The paper also includes interesting analyses of learned group structure (isolated agents, algebraic connectivity) that many MARL readers will find valuable.

**Claims And Evidence:**

No

**Claims Explanation:**

The paper provides convincing evidence that IA-KRC improves performance in the specific SMACv2 variants and self-play protocol studied (including ablations showing the contributions of both K-step reachability and interference modeling). However, several headline claims are broader than what is cleanly supported, notably the “state-of-the-art” claim in the abstract.

More importantly, the evidence is hard to fully trust without a precise accounting of what information is available during centralized training vs decentralized execution/evaluation (e.g., map layers, enemy attributes, ground truth attack intent labels/signals). The method’s interference model seems to explicitly depend on attributes like health/attack power and a supervised signal tied to actual trajectories, and key details for the multi-layer map are deferred to the appendix.

Together, this leaves ambiguity about whether the improvements would hold under standard Dec-POMDP constraints and truly decentralized evaluation.

**Requested Changes:**

### Major
- Explicitly list the information available to each agent (and any centralized component) during training and during decentralized execution/evaluation, including: what is in each map layer; what is observed vs inferred; and whether any enemy attributes (e.g., health/attack power) are directly accessible.

- Clarify decentralized execution details: how groups/leaders are formed online (frequency, protocol, what messages are exchanged, and how a follower selects among candidate leaders). The described leader election mechanism seems to require global information, which makes it unclear how each agent's leader-message input is obtained during decentralised execution.

- Clarify evaluation: whether FW/HW/FL are measured under frozen policies and decentralized rollouts, and how self-play online updates affect reported "cumulative win rates". Also why is the performance of the baselines in the standard unmodified 8m task in Figure d so poor, and significantly different from the ones reported in prior works [1]

### Minor

- Tone down or qualify “state-of-the-art” claims (or expand evaluation) to better match the scope (custom SMACv2 maps + self-play).
- Disambiguate notation around “state”. Is it the global Dec-POMDP state as introduced in the background, or the agent-local “state” (position) used in reachability distance computations?
-  Discuss K selection and practicality. K is set to 9 in the experiments and ablated from 3 to 12, but it is unclear how those choices are related to the map grid-size or intricacies of the specific chosen benchmark (SMACv2). These would help to provide guidance/heuristics for choosing K and sensitivity across maps and other domains.


[1] Hu, Shengchao, et al. "Communication learning in multi-agent systems from graph modeling perspective." IEEE Transactions on Knowledge and Data Engineering (2026).

---

> ### Author Response · Authors · 2026-02-09
> **Rebuttal to Reviewer yEr1(Experimental Results)(1/2)**
>
> ## Explanation Regarding Standard 8m Environment Experimental Results
>
> Thank you very much for carefully pointing out the abnormal performance of baseline methods in Figure 6(d). After thorough code review and problem investigation, we discovered a critical configuration error **affecting the standard 8m test environment**. We explain this in detail below:
>
> ### Root Cause
>
> In the standard SMAC experimental configuration, our units should use Marine_RL (unit type ID=1939), which is a special unit type defined in the SMAC StarCraft II mod that **disables all automatic AI behaviors at the unit definition level**, ensuring units are entirely controlled by reinforcement learning policies; while enemy units use standard Marine (unit type ID=48), controlled by StarCraft II's built-in AI.
>
> However, in our implementation (`starcraft2custom.py` line 1968), due to incorrectly using the `custom=False` parameter, our units were created as standard Marine (ID=48) instead of Marine_RL (ID=1939). This error caused **dual control conflict**: our units simultaneously received automatic attack commands from the built-in AI and control commands from the reinforcement learning policy. The conflict between these two control sources resulted in highly chaotic unit behavior, severely impairing the training and execution effects of all algorithms.
>
> Specifically:
>
> - **Expected configuration**: Our side uses ID=1939 (pure RL control), enemy uses ID=48 (built-in AI control)
> - **Actual configuration**: Our side uses ID=48 (RL + built-in AI conflict), enemy uses ID=48 (built-in AI control)
> - **Impact**: All algorithms performed abnormally poorly in Figure 6(d), which is a systematic problem caused by environment configuration errors, not performance issues of the algorithms themselves
>
> ### Technical Explanation of IA-KRC's Relative Robustness
>
> It is worth noting that under this abnormal configuration, IA-KRC's performance degradation was relatively smaller (about 80%), while other baseline methods performed worse. We believe this may be related to IA-KRC's design characteristics:
>
> 1. **Locality of K-step reachability constraints**: IA-KRC restricts communication to local neighborhoods through K-step reachability constraints. This locality design naturally limits the impact range of abnormal behaviors, giving the system a certain fault tolerance.
> 2. **Leader-follower grouping architecture**: IA-KRC adopts a grouped cooperation mechanism where agents are divided into multiple independent cooperation groups, each consisting of one leader and several followers. This structured organization may have partially mitigated the spread of global control conflicts.
>
> ### Validity of Self-Play Experiments
>
> **Important Note**: After rigorous review, we confirm that this configuration error **only exists in the standard 8m environment testing** and does not affect the validity of self-play experiments (Dense-Obstacle and Maze-Structure scenarios) at all.
>
> In the self-play experiments, both sides correctly used the ID=1939 (Marine_RL) unit type (through the `custom=True` parameter), therefore:
>
> - Both sides have completely symmetric conditions, with no unit type inconsistency issues
> - All units are purely controlled by RL policies, without AI behavior interference
> - IA-KRC's performance advantages over all baseline methods were obtained under fair and correct experimental conditions
> - All comparative experimental results under the self-play framework (including win rates, convergence speed, organizational structure analysis, etc.) are genuinely valid

---

> ### Author Response · Authors · 2026-02-09
> **Rebuttal to Reviewer yEr1(Experimental Results)(2/2)**
>
> ### Corrective Measures
>
> We have fixed the code error and re-ran the 8m environment experiments under the standard SMAC configuration (all parameters use default parameters from the authors' open-source code). In the revised manuscript, we will:
>
> 1. **Update Figure 6(d)** to show the corrected experimental results
> 2. **Update Section 5.4** to supplement detailed discussion of performance under correct configuration
>
> **Corrected experimental results (see Section 5.4 in the original text)**: Under correct configuration, IA-KRC maintains clear advantages in the standard 8m obstacle-free environment, achieving faster convergence speed and attaining a win rate second only to CommFormer. CommFormer's training duration exceeds IA-KRC by more than 4 times, while IA-KRC's training time only increases by approximately 19% compared to other baseline algorithms, demonstrating that IA-KRC achieves a better balance between computational efficiency and performance.
>
> These results fully confirm that: even in simplified environments with significantly reduced communication complexity, IA-KRC still maintains advantages, indicating that its advantages stem not only from geometric constraints but more importantly from the interference-aware modeling mechanism. The dynamic influence map enables agents to effectively capture congested and conflict areas as well as temporal instabilities among teammates—these dynamic interference factors significantly impact cooperation even in environments without physical obstacles. As stated in the original text, IA-KRC's core mechanisms—reachability filtering and interference prediction—can effectively generalize beyond complex topologies, providing robust and significant performance advantages even in environments with minimal structural constraints.
>
> We sincerely apologize for this oversight, but emphasize that **this configuration error does not affect the validity of the paper's core contributions**: all experiments under the self-play scenarios were completed under correct configuration, and IA-KRC's advantages in complex topologies and dynamic interference environments have been fully validated.

---

> ### Author Response · Authors · 2026-02-09
> **Rebuttal to Reviewer yEr1(1/7)**
>
> ## Point 1: Information Availability and Dec-POMDP Compliance
>
> **Theme: Information Availability and Dec-POMDP Compliance**
>
> ### Related Reviewer Concerns
>
> **Weakness 1:**
>
> > "State-of-the-art framing is too strong given the evaluation protocol/domain scope (custom SMACv2 + self-play) and unclear assumptions behind the Dec-POMDP setup."
>
> **Weakness 2:**
>
> > "Privileged/domain-specific information seems heavy (and not clearly enumerated): IA-KRC's cost model uses agent attributes like health/attack power, and the interference direction is trained via supervised learning to match an agent's actual trajectory. The multi-layer map claims it can compute shortest transition distances "between any pair of states at any time," but key implementation details are deferred to the appendix. It is also unclear if the notation s refers to the agent position or environment state (in which case I assume it includes more than the agent x,y position in SMAC)."
>
> **Major Change 1:**
>
> > "Explicitly list the information available to each agent (and any centralized component) during training and during decentralized execution/evaluation, including: what is in each map layer; what is observed vs inferred; and whether any enemy attributes (e.g., health/attack power) are directly accessible."
>
> **Evidence Support Issue:**
>
> ---
>
> ### Response
>
> Thank you for your attention to the implementation details of the IA-KRC algorithm. We first clarify: **All information usage in IA-KRC strictly follows CTDE (Centralized Training, Decentralized Execution) and Dec-POMDP specifications, without introducing any privileged information**. Regarding the detailed implementation of the algorithm, **most technical details have been fully described in the Appendix** (including multi-layer map algorithms, interference prediction modules, neural network architectures, etc.). To better address your concerns, we will supplement the main text of the revised version with necessary information availability explanations, enabling readers to clearly understand IA-KRC's information usage at various stages of training and execution.
>
> Below, we provide detailed answers to your specific questions point by point:
>
> #### 1.1 Clarification of Dec-POMDP Assumptions
>
> IA-KRC follows the standard Dec-POMDP framework and CTDE paradigm. Main assumptions include: during training, access to global state $s$ is allowed (mixer network), while during execution only local observation $o_i$ is used; limited communication is allowed (3-dimensional message vector, updated every 4 steps); vision range constraints (sight_range=9); symmetric information access for both sides. These assumptions comply with standard CTDE practices.
>
> #### 1.2 Information Accessible to Each Agent During Training Phase
>
> During the training phase, each agent accesses two types of information, complying with the standard CTDE paradigm:
>
> (1) **When executing policy**: Local entity observation $o_i$ (containing position, health, type of allies/enemies within vision range), own action history, received leader message (3-dimensional vector), RNN hidden state
>
> (2) **During training optimization**: Historical experiences in replay buffer (for off-policy learning), global state $s$ used by mixer network (only for value decomposition, does not affect agent policy)
>
> #### 1.3 Information Accessible to Centralized Components During Training Phase
>
> Centralized components and their information access:
>
> (1) **QMIX Mixer**: Global state $s$ (generating mixing weights through hypernetwork), combining individual $Q_i$ into $Q_{tot}^g$
>
> (2) **Controller**: Collects all agents' $N_i^{(K)}$ scores for leader election, synchronizes grouping results to agents
>
> (3) **Replay Buffer**: Stores all agents' $(s, \mathbf{a}, r, s')$ experiences for mini-batch sampling
>
> (4) **Intent Network Trainer**: Accesses enemy trajectories in historical replay buffer for supervised learning
>
> #### 1.4 Information Accessible to Each Agent During Decentralized Execution/Evaluation Phase
>
> The execution phase strictly follows Dec-POMDP constraints, with each agent only accessing:
>
> (1) **Entity observation**: Information about all units within vision range (position, health/shield, unit type), as well as allies' available actions
>
> (2) **Self information**: Action history and RNN hidden state
>
> (3) **Communication information**: Message vector from leader and grouping information (updated every msg_T=4 steps)
>
> (4) **Local map**: Obstacle observations within vision (LOS) and own movement feedback
>
> Cannot access: Global state $s$, information outside vision range, other agents' internal states, future information

---

> ### Author Response · Authors · 2026-02-09
> **Rebuttal to Reviewer yEr1(2/7)**
>
> #### 1.5 What Information Is Contained in Each Map Layer
>
> (See Appendix Section A.2 for details)
>
> Content and information sources for the three map layers:
>
> **Geometry Layer**: Grid representation of physical obstacles. Information source is LOS observations within agent's vision range, incrementally updating obstacle and free cells through `update_from_sight()`
>
> **Regulation Layer**: Directed graph of environmental rule constraints (doors, traffic rules, etc.). Information source is agent's own $(s_t, a_t, s_{t+1}, success)$ records, updating edges based on success/failure of action transitions
>
> **Interference Layer**: Potential field representation of enemy interference. Information source is observations of enemy entities **within vision range** (position, health, trajectory), computing grid-based cost map $M_{cost}$
>
> All layers are based on local observations and self feedback, requiring no global information or privileged interfaces
>
> #### 1.6 What Information Is Observed vs Inferred
>
> (Inference methods detailed in Appendix Section A.3)
>
> **Direct observation** (obtained from SMAC environment): entity positions/health/shield/types, obstacle positions (LOS), action execution results, rewards
>
> **Inferred/computed** (based on observations):
>
> - Attack intent angle $\theta_e$: Inferred through trained MLP from enemy's current state (position + trajectory + health)
> - Threat level $T_e$: Heuristically computed from observed trajectory, attack history, health
> - Reachability $\mathcal{S}_{IA}(s, K)$: Computed by running Dijkstra's algorithm on multi-layer map
>
> #### 1.7 Are Enemy Attributes Directly Accessible
>
> Yes, directly accessible under SMAC standard configuration. The SMAC environment default setting is `obs_all_health=True`, allowing agents to observe health and shield of all units (including enemies) **within vision range** as standard fields of entity observation. Attack power is implicit through unit type (different types have fixed attack power), and unit type is encoded through type bits in entity observation. These are all components of SMAC's standard entity observation, not privileged information.
>
> **Regarding the method's generality and modular design**: It should be emphasized that the IA-KRC framework does not depend on privileged information provided by specific environments. The Interference Layer adopts a **modular design**, whose input interface (enemy attributes such as health, position, trajectory) can be obtained through two approaches:
>
> (1) **Direct observation**: In supported environments (such as SMAC with `obs_all_health=True`), directly obtained from entity observation;
>
> (2) **Opponent modeling inference**: In environments that do not provide enemy attributes, these states can be estimated through mature opponent modeling methods, such as using historical trajectory prediction, inverse reinforcement learning, or neural network modeling techniques.
>
> This modular design ensures that IA-KRC's core mechanism (K-step reachability + interference-aware selection) does not depend on specific information acquisition methods, providing broad cross-environment applicability. The Interference Layer only defines the **interface specification** for required information, not the **acquisition method**, allowing the framework to flexibly adapt to different MARL environments and information constraint conditions.

---

> ### Author Response · Authors · 2026-02-09
> **Rebuttal to Reviewer yEr1(3/7)**
>
> #### 1.8 Specific Information Used
>
> (Threat level calculation detailed in Appendix Section A.3.1)
>
> Enemy attributes used by the Interference Prediction Module include:
>
> (1) **Health** (normalized health value)
>
> (2) **Recent trajectory** (recent trajectory, used to calculate movement threat)
>
> (3) **Attack history** (attack history, used to calculate attack threat)
>
> (4) **Position** (position information)
>
> All this information comes from SMAC standard entity observation. The environment default setting is `obs_all_health=True`, allowing agents to observe the health/shield of all units within vision range. Position and trajectory are obtained from the entity's global coords field. All information is obtained through entity information within visual observation range, without requiring privileged interfaces.
>
> #### 1.9 Does Supervised Learning Use Privileged Information
>
> (Training process detailed in Appendix Section A.3.1)
>
> No, it is not privileged information. Supervised learning follows the standard offline training + online inference mode:
>
> **Training phase**: The network uses historical data from the replay buffer, with $(state_t, movement_{t+1})$ pairs as training samples. At this point, $movement_{t+1}$ has already occurred and been recorded, representing known historical information, not future privileged information.
>
> **Execution phase**: The network has been trained, taking enemy's current state (position, trajectory, health) as input and outputting predicted attack intent vector (2D vector). Only current observations are needed for inference, without accessing future information.
>
> This is similar to opponent modeling methods in MARL, complying with the standard practice in the CTDE paradigm of "learning models during training, using models during execution."
>
> #### 1.10 Implementation Details of Multi-layer Map
>
> (See Appendix Section A.2 for details)
>
> "Computing at any time" means being able to query distances at any moment during the training/execution process, implemented through incremental updates rather than global precomputation.
>
> Information sources for the three map layers:
>
> **Geometry Layer**: Uses LOS observations within agent's vision range, incrementally updating obstacle and free cells through `update_from_sight()`
>
> **Regulation Layer**: Uses agent's own $(s_t, a_t, s_{t+1}, success)$ records, updating directed edges based on success/failure of action transitions
>
> **Interference Layer**: Uses observations of enemy entities within vision range (position, health, trajectory), computing grid-based cost map
>
> All layers are based on local observations and self action feedback, complying with Dec-POMDP information constraints. During execution, only local observations are needed to maintain and query the map, without requiring global state or privileged information. Detailed algorithms in Appendix.
>
> ---

---

> ### Author Response · Authors · 2026-02-09
> **Rebuttal to Reviewer yEr1(4/7)**
>
> ## Point 2: Decentralized Execution Details
> **Theme: Decentralized Execution Details**
> ### Related Reviewer Concerns
> **Major Change 2:**
> > "Clarify decentralized execution details: how groups/leaders are formed online (frequency, protocol, what messages are exchanged, and how a follower selects among candidate leaders). The described leader election mechanism seems to require global information, which makes it unclear how each agent's leader-message input is obtained during decentralised execution."
> ### Response
> #### 2.1 How Groups/Leaders Are Formed Online
>
> IA-KRC follows the standard CTDE paradigm for grouping. The leader election process includes three steps:
>
> (1) Each agent i computes the number of reachable neighbors within its K-step reachable domain based on local information: $N_i^{(K)} = |\{j \in \mathcal{N}: d_{IA}(s_i, s_j) \leq K\}|$
>
> (2) The training framework collects all agents' $N_i^{(K)}$ scores and selects the top M agents with the highest scores as leaders
>
> (3) Synchronizes grouping results to all agents
>
> #### 2.2 Formation Frequency
>
> Reformed every 4 time steps, determined by parameter msg_T=4, consistent with the default settings of the SOG method.
>
> #### 2.3 What Protocol Is Used
>
> IA-KRC's protocol contains two levels, fully consistent with the MARL framework described in Section 4.3:
>
> **(1) Grouping protocol** (executed every msg_T steps):
>
> - Compute each agent's $N_i^{(K)}$ (K-step reachable neighbor count)
> - Select top-M as leaders
> - Followers select leaders based on K-step reachability and load balancing
>
> **(2) Intra-group cooperation protocol**:
>
> - Agents within groups learn cooperation strategies through the QMIX framework
> - Each group maintains joint action-observation history $\tau_g$
> - Training through minimizing group joint TD loss
>
> #### 2.4 What Messages Are Exchanged
>
> The messages exchanged are low-dimensional summary vectors of observation information (3-dimensional continuous vectors). Each agent encodes its local observation into a 3-dimensional vector through a neural network. Followers within groups send individual messages to the leader, who uses average pooling to aggregate and then broadcasts to all members within the group. Agents concatenate received messages with RNN hidden state when computing Q-values.
>
> #### 2.5 How Followers Select Among Candidate Leaders
>
> Follower selection follows two constraints:
>
> (1) **Reachability constraint**: Followers only consider leaders within their K-step reachable domain as candidates
>
> (2) **Load balancing**: From the candidate set, select the leader with the smallest current group size
>
> #### 2.6 Consistency Between Leader Election Mechanism and Decentralized Execution
>
> IA-KRC follows the standard CTDE paradigm in MARL. Leader election, as a communication topology organization mechanism, is centrally executed by a controller during both training and execution (collecting all agents' $N_i^{(K)}$ values and selecting top-M every msg_T steps), then synchronizing the grouping results (M leader IDs and follower assignments) to all agents in a lightweight manner. This does not violate the core constraints of Dec-POMDP because:
>
> **(1) Policy decision level is fully decentralized**: Each agent only relies on local observation $o_i$, RNN hidden state, and received messages when selecting actions, without accessing global state or other agents' internal states.
>
> **(2) Communication topology organization allows lightweight coordination**: Grouping information (M leader IDs + follower assignments) is fixed-format lightweight data (total communication volume is O(N)), synchronized through standard communication protocols. This is similar to QMIX where the mixer uses global state for value decomposition—the key is that global information does not directly participate in agents' action selection but is used to assist training or communication organization.
>
> **(3) Flexibility in actual deployment**: While our experiments use a centralized coordinator for grouping (which is feasible in many actual MARL deployment scenarios), if fully decentralized deployment is needed, it can be achieved through consensus protocols (such as each agent broadcasting $N_i^{(K)}$ and independently computing top-M), and our core contribution—partner selection criteria based on K-step reachability and interference-awareness—remains unchanged.
>
> #### 2.7 How Each Agent Obtains Leader Message Input
>
> During the decentralized execution phase, message passing is entirely achieved through local communication, without requiring global information:
>
> (1) **Message encoding**: Each agent encodes its local observation into a low-dimensional message vector (3-dimensional)
>
> (2) **Intra-group aggregation**: Followers send messages to their assigned leader, who uses average pooling to aggregate intra-group messages
>
> (3) **Message reception**: Leader broadcasts aggregated message to group members, and each agent concatenates received message with RNN hidden state for Q-value computation

---

> ### Author Response · Authors · 2026-02-09
> **Rebuttal to Reviewer yEr1(5/7)**
>
> ## Point 3: Evaluation Protocol Clarification
>
> **Theme: Evaluation Protocol Clarification**
>
> ### Related Reviewer Concerns
>
> **Weakness 3:**
>
> > "Evaluation protocol ambiguity: results are reported under a self-play setup where both sides update online. This is a nice idea given the map modifications, but the unexpected poor performance of the baselines in the standard unmodified 8m task in Figure d (relative to reported results in prior works) makes all results questionable. It is also unclear whether FW/HW/FL are computed from decentralized evaluation with frozen policies or from the online learning process (with centralised information). The paper notes "all metrics refer to IA-KRC" but doesn't fully resolve interpretability."
>
> **Major Change 3:**
>
> > "Clarify evaluation: whether FW/HW/FL are measured under frozen policies and decentralized rollouts, and how self-play online updates affect reported "cumulative win rates". Also why is the performance of the baselines in the standard unmodified 8m task in Figure d so poor, and significantly different from the ones reported in prior works [1]"
>
> ---
>
> ### Response
>
> #### 3.1 Necessity and Credibility of Self-Play Setup
>
> Self-play online updates are a necessary choice for validating performance in complex topological environments. Built-in AI does not support obstacle occlusion (vision/attack are not blocked by obstacles) and is closed-source with rules that cannot be modified, making it impossible to validate cooperation performance under complex topologies.
>
> **Fairness guarantee**: Symmetric map design, same environmental rules for both sides, randomized initial positions.
>
> **Credibility guarantee**: Although both sides are updating, sustained win rate advantages reflect differences in learning efficiency and policy quality of algorithms. Result credibility is verified through: (1) 5 random seed repeated experiments, (2) 2.0M steps long-term trends rather than short-term fluctuations, (3) consistent advantages over multiple baselines. Similar to self-play systems like AlphaGo/AlphaStar, sustained win rate differences indicate algorithmic superiority.
>
> #### 3.2 Calculation Method for FW/HW/FL
>
> Directly collected during the online learning process, which is standard practice for self-play scenarios. Specific method:
>
> During training, after each episode ends, win/loss results are recorded (determined based on surviving agent count), and wins/losses/draws are accumulated. Every 10,000 steps, the current cumulative win rate is recorded (`unit_win_rate = unit_battles_won / unit_battles_game`). FW (final win rate) is the cumulative win rate at the end of 2.0M steps of training, HW (highest win rate) is the peak recorded during the entire training process and corresponding steps.
>
> **Reasons for adopting online statistics**: (1) Self-play requires high sample efficiency; frozen evaluation would double computational cost and waste samples, (2) This is standard practice for self-play scenarios (AlphaGo, AlphaStar, etc.), (3) Cumulative statistics combined with long-term trends (2.0M steps) and multiple seed repetitions (5 seeds) have sufficiently verified stability.
>
> **Note**: Although online statistics include exploration noise (ε-greedy), long-term cumulative trends over 2.0M steps can effectively smooth short-term fluctuations, reflecting true policy quality differences.
>
> #### 3.3 Whether Centralized Information Is Used During Evaluation
>
> Execution level is fully decentralized, statistical level records centrally after episode ends.
>
> **Execution level** (decentralized, complying with Dec-POMDP): Agents only use local observations when selecting actions, communication follows bandwidth limitations, grouping based on local reachability information, fully complying with Dec-POMDP constraints.
>
> **Statistical level** (after episode ends): Environment determines win/loss based on surviving agent count of both sides (ally_alive vs enemy_alive), recording to cumulative statistics. This does not violate Dec-POMDP because statistics occur after episode termination and do not affect agents' online decisions.
>
> #### 3.4 Meaning of "all metrics refer to IA-KRC"
>
> This indicates that FW/HW/FL reported in the Table are all collected from IA-KRC's perspective.
>
> **Specific meaning**: Each row in the Table represents self-play results of IA-KRC vs corresponding baseline. For example, "DPP, FW=88.37%, FL=2.8%" means: In the confrontation of IA-KRC vs DPP, IA-KRC won 88.37% of episodes, lost 2.8%, with the remaining approximately 8.8% being draws.
>
> **Symmetry**: Under symmetric maps, IA-KRC's win% + draw% + loss% = 100%. Reporting FW and FL allows readers to fully understand the win/loss distribution.
>
> ---

---

> ### Author Response · Authors · 2026-02-09
> **Rebuttal to Reviewer yEr1(6/7)**
>
> ## Point 4: Minor Revisions (SOTA Claims & Notation)
>
> **Theme: Minor Revisions (SOTA Claims & Notation Disambiguation)**
>
> ### Related Reviewer Concerns
>
> **Weakness 1 (Regarding SOTA claims):**
>
> > "State-of-the-art framing is too strong given the evaluation protocol/domain scope (custom SMACv2 + self-play) and unclear assumptions behind the Dec-POMDP setup. The abstract explicitly claims "superior performance compared to state-of-the-art baselines.""
>
> **Weakness 2 (Regarding notation ambiguity):**
>
> > "It is also unclear if the notation s refers to the agent position or environment state (in which case I assume it includes more than the agent x,y position in SMAC)."
>
> **Minor Change 1:**
>
> > "Tone down or qualify "state-of-the-art" claims (or expand evaluation) to better match the scope (custom SMACv2 maps + self-play)."
>
> **Minor Change 2:**
>
> > "Disambiguate notation around "state". Is it the global Dec-POMDP state as introduced in the background, or the agent-local "state" (position) used in reachability distance computations?"
>
> **Evidence Support Issue:**
>
> > "The paper provides convincing evidence that IA-KRC improves performance in the specific SMACv2 variants and self-play protocol studied (including ablations showing the contributions of both K-step reachability and interference modeling). However, several headline claims are broader than what is cleanly supported, notably the "state-of-the-art" claim in the abstract."
>
> ---
>
> ### Response
>
> #### 4.1 Clarification of SOTA Statements
>
> Thank you for your concern about the evaluation scope. We believe the "state-of-the-art" statement is reasonable, mainly based on the following considerations:
>
> **Consistency with domain standard benchmarks**: SMAC (StarCraft Multi-Agent Challenge) is the currently recognized gold standard testing platform in the field of cooperative multi-agent reinforcement learning. Many widely recognized SOTA algorithms, such as **QMIX** [1], **QPLEX** [2], and **RODE** [3], have their core performance claims and SOTA status primarily established on evaluations using the SMAC benchmark. We achieved significantly superior performance compared to these baselines in various complex scenarios of SMACv2 (which is more challenging than SMAC).
>
> [1] Rashid, T., Samvelyan, M., Schroeder de Witt, C., Farquhar, G., Foerster, J., and Whiteson, S. QMIX: Monotonic Value Function Factorisation for Deep Multi-Agent Reinforcement Learning. In ICML, 2018.
>
> [2] Wang, J., Ren, Z., Liu, T., Yu, Y., and Zhang, C. QPLEX: Duplex Dueling Multi-Agent Q-Learning. In ICLR, 2021.
>
> [3] Wang, T., Gupta, T., Mahajan, A., Peng, B., Whiteson, S., and Zhang, C. RODE: Learning Roles to Decompose Multi-Agent Tasks. In ICLR, 2021.
>
> #### 4.2 Notation Disambiguation
>
> We understand your concern. In the MARL field, $s$ is typically used to represent global state. The notation $s$ in the paper has two uses, which may cause confusion:
>
> **Global Dec-POMDP state** (Preliminaries): $s \in \mathcal{S}$ represents the global state containing complete information about all agents and the environment
>
> **Agent local state** (Definitions 1-3): $s_1, s_2 \in S$ in reachability definitions refers to agent's local state (containing position information but not equivalent to global state)
>
> **Revision plan**: To follow field conventions and eliminate ambiguity, we modify the notation representing agent local state from $s$ to $x$ in the Method chapter (Definitions 1-3 and related formulas), and use $\mathcal{X}$ to represent the agent state space. This clearly distinguishes two levels of information:
>
> - Global state $s \in \mathcal{S}$: Contains complete information about all agents and environment (only used for value decomposition during training)
> - Local state $x_i \in \mathcal{X}$: State representation extracted from observation $o_i$ (such as position), used for reachability computation
>
> ---

---

> ### Author Response · Authors · 2026-02-09
> **Rebuttal to Reviewer yEr1(7/7)**
>
> ## Point 5: K Selection and Practicality
>
> **Theme: Practicality of K Value Selection**
>
> ### Related Reviewer Concerns
>
> **Minor Change 3:**
>
> > "Discuss K selection and practicality. K is set to 9 in the experiments and ablated from 3 to 12, but it is unclear how those choices are related to the map grid-size or intricacies of the specific chosen benchmark (SMACv2). These would help to provide guidance/heuristics for choosing K and sensitivity across maps and other domains."
>
> ---
>
> ### Response
>
> #### 5.1 Basis for Choosing K=9
>
> The choice of K=9 is based on two considerations:
>
> (1) **Fair comparison with baselines**: In the SMAC environment, sight_range=9. Choosing K=9 makes K-step reachability comparable to the perception range of vision-based methods, ensuring fair comparison.
>
> (2) **Ablation experiment validation** (Table 2): Testing K=3,6,9,12 four settings, results show K=9 achieves optimal performance (83.63%), superior to K=3 (75.48%), K=6 (79.89%), and K=12 (73.66%). Analysis shows K=9 maximizes medium-range partner selection without excessively extending prediction uncertainty.
>
> #### 5.2 Relationship Between K Value and Map Grid Size
>
> K value needs to consider map scale and agent dispersion:
>
> **Relative distance principle**: K should cover the spatial range needed for effective cooperation. In SMACv2's 32×32 map, K=9 can cover about 28% of the map width, sufficient to connect dispersed teammates without excessive diffusion.
>
> **Group density impact**: In agent-dense areas, smaller K can connect sufficient cooperators; in sparse distributions, larger K is needed to avoid isolated agents. Our experiments show that when the K selection to "average inter-agent distance" ratio is about 2-3 times, the effect is better.
>
> #### 5.3 Relationship Between K Value and Benchmark Task Characteristics
>
> K value is related to task cooperation tightness and terrain complexity:
>
> **Terrain complexity**: In complex terrains like Dense-Obstacle and Maze, K=9 can traverse obstacles to connect reachable teammates. Ablation experiments show K too small (such as 3) limits cooperation range, K too large (such as 12) introduces noise accumulation.
>
> **Cooperation requirements**: SMACv2 combat tasks require medium-range cooperation (focus fire, flanking), and the range covered by K=9 matches these tactical needs. In tasks with different cooperation modes (such as escort, defense), optimal K may differ.
>
> #### 5.4 How to Select K for New Tasks/Maps
>
> Heuristic method for selecting K through rapid pre-experiments is recommended:
>
> (1) **Initial range**: Set K in the interval [vision range×0.5, vision range×1.5]
>
> (2) **Pre-training testing**: Test 3-4 candidate K values with a small number of training steps (such as 50k steps)
>
> (3) **Evaluation metrics**: Observe isolated agent ratio and $\lambda_2$ (algebraic connectivity). Select K that minimizes isolated agents and has higher $\lambda_2$
>
> (4) **Fine-tuning**: Perform fine-grained search (±1-2 steps) near the selected K

---

> ### Author Response · Authors · 2026-02-28
> **Follow-up for Reviewer yEr1**
>
> **Dear Reviewer yEr1,**
>
> Thank you again for the time you have dedicated to our paper and for your valuable suggestions for improvement.
>
> We would like to follow up on the rebuttal we submitted and the revised PDF of the manuscript. We sincerely hope that our detailed responses, along with the corresponding updates made to the paper, have effectively addressed your concerns.
>
> If you have any further questions or require additional clarification regarding our responses, please do not hesitate to let us know. We would be more than happy to provide any further information to ensure that all points have been addressed to your satisfaction.
>
> Thank you once again for your hard work and guidance.
>
> **Best regards,**
>
> **The Authors**

---

### Author Response · Authors · 2026-02-10
**Minor revision of paper**

# Revision Summary

Based on the reviewers' feedback and our careful reassessment of the experimental results and paper structure, we have made several important revisions. The main changes include:

## Part I: Restructuring of Chapter 4 (Method)

Based on the reviewers' feedback and our reassessment of the paper structure, we have restructured Chapter 4 (Method). The main revisions include:

**Reorganization of Section Structure:** We have restructured the Method section from subsections 4.1 and 4.2 to 4.1, 4.2, and 4.3. The specific revisions made to each section are as follows:

**In Section 4.1 (Computing K-step reachability using multi-layer map):** We added in-depth analysis addressing the novelty and technical challenges in extending K-step reachability from single-agent to multi-agent settings.

**In Section 4.2 (Computing Cooperation Cost with an Interference Potential Field):** We enhanced the explanation and justification for choosing the interference prediction module.

**In Section 4.3 (MARL with IA-KRC):** We clarified that the leader-follower framework in our algorithm serves solely for clustering and grouping purposes, while the within-group training employs QMIX value decomposition.

We will provide detailed rebuttal responses to each reviewer shortly.

## Part II: Revisions to Experimental Results and Notation

**1. Correction of Figure 5(d)**

**2. Restructuring of Section 5.4 (Generalization to Standard Obstacle-Free Environments)**

**3. Notation Disambiguation for Local State**

**4. Clarification of Grouping Structure Description**

---

---

### Author Response · Authors · 2026-03-03
**Follow-up for the paper**

Dear Action Editor

We hope this message finds you well. We are writing to respectfully follow up
on the current reviewing status of our submission.

Since submitting our revised manuscript and detailed rebuttal responses on
February 9–10, 2026, we have been actively engaging with all three reviewers.
We are pleased to note that Reviewer cdvX has already expressed satisfaction
with our responses on February 12, and we sent follow-up messages to Reviewers
yEr1 and ERuu on February 28 to address any remaining questions.

It has now been nearly four weeks since we submitted our revisions. Given that
the review process has been ongoing for some time, we would be most grateful if
you could kindly consider moving toward a final editorial decision at your
earliest convenience. We fully understand that your time is valuable and that
the review process requires careful consideration, and we sincerely apologize
for any inconvenience this follow-up may cause.

Should any additional information or clarification be needed to facilitate your
decision, we remain readily available.

Thank you very much for your continued effort and guidance throughout this
process.

Best regards,

The Authors

---

> ### Author Response · Authors · 2026-03-15
> **Follow-up Inquiry on the Review Progress of Our Submitted Manuscript**
>
> Dear Action Editor,
>
> We are writing to politely follow up on the current review progress of our submitted manuscript.
>
> As outlined in the official email notification we received from TMLR, all reviewers were expected to submit their final review comments by March 2, 2026, and a final editorial decision will be issued within one week after all complete final reviewer comments are received. With this in mind, we are reaching out to kindly inquire about the current review progress of our submission.
>
> We fully understand that you have an extremely busy schedule with your academic and editorial responsibilities, and we well appreciate that there may be unforeseen delays throughout the review process. We sincerely apologize for any inconvenience this inquiry may cause, and deeply appreciate the time and effort you have dedicated to handling our manuscript.
>
> Best regards,
>
> The Authors

---

### Decision · Action_Editor_LX81 · 2026-04-03

**Recommendation:** Accept as is

**Audience:**

Yes

**Audience Explanation:**

The work can be of interest to Multi-agent RL subcommunity in TMLR.

**Claims And Evidence:**

Yes

**Claims Explanation:**

The paper mainly combines K-step reachability constraints with an interference prediction module to guide communication partner selection. The reviewers think that the design is well motivated and empirically validated. Although the contributions are somewhat narrow in scope (experiments are conducted within variants of SMAC and related environments) and the proposed approach heavily depends on a per-environment tuned K-step hyperparameter, the reviewers are satisfied with the updated draft and the provided clarifications in the rebuttal.